# 1,2,4,5-Tetrazine-tethered probes for fluorogenically imaging superoxide in live cells with ultrahigh specificity

Xuefeng Jiang[1,6], Min Li [1,6], Yule Wang [1], Chao Wang [2], Yingchao Wang [1,3], Tianruo Shen[2], Lili Shen[1], Xiaogang Liu [2], Yi Wang [1,3,4,5] ✉ & Xin Li [1,5] ✉

Superoxide ($O_2^{·-}$) is the primary reactive oxygen species in mammal cells. Detecting superoxide is crucial for understanding redox signaling but remains challenging. Herein, we introduce a class of activity-based sensing probes. The probes utilize 1,2,4,5-tetrazine as a superoxide-responsive trigger, which can be modularly tethered to various fluorophores to tune probe sensitivity and emission color. These probes afford ultra-specific and ultra-fluorogenic responses towards superoxide, and enable multiplexed imaging of various cellular superoxide levels in an organelle-resolved way. Notably, the probes reveal the aberrant superoxide generation in the pathology of myocardial ischemia/reperfusion injury, and facilitate the establishment of a high-content screening pipeline for mediators of superoxide homeostasis. One such identified mediator, coprostanone, is shown to effectively ameliorating oxidative stress-induced injury in mice with myocardial ischemia/reperfusion injury. Collectively, these results showcase the potential of 1,2,4,5-tetrazine-tethered probes as versatile tools to monitor superoxide in a range of pathophysiological settings.

Reactive oxygen species (ROS) include a broad variety of oxygen derivatives that are labile, oxidative, and short-lived[1]. Their similar structures and reactivity make the specific detection of a single ROS highly challenging. Yet, to accurately elucidate the molecular mechanism underlying the pleiotropic roles of various ROS in physiology and pathology[2], and to translate redox signaling into effective therapies, it is essential to develop robust tools that specifically detect a ROS of interest while being silent towards others[3–6], ideally in live cells with desirable spatiotemporal resolution.

Probably the most important ROS is superoxide ($O_2^{·-}$). Superoxide is generated by a variety of enzymes, such as the mitochondrial electron transport chain (ETC) and NADPH oxidases (NOX)[7,8]. Its dismutation by superoxide dismutase (SOD) yields $H_2O_2$ and fuels other

ROS for redox signaling (Fig. 1A)[9–12]. Imbalanced superoxide homeostasis initiates oxidative stress which is implicated in the etiology of aging[13], cancer[14], neurodegenerative diseases[15], diabetes[16], cardiovascular diseases[17], etc. Given the pivotal roles of superoxide in redox signaling and redox homeostasis, the specific detection of superoxide has long been a hot topic attracting continuous research efforts[18].

Various methods have been developed to detect superoxide, including electron paramagnetic resonance (EPR) spin-trapping technique[19], electrochemical sensors[20], spectrophotometric assays[21], chemiluminescent assays[22], and fluorescent imaging[23]. Among these diverse methods, fluorescent imaging is the most desirable, because it is compatible with live cells and permits the direct in situ tracing of superoxide dynamics with unprecedented spatiotemporal resolution.

[1]College of Pharmaceutical Sciences, Zhejiang University, 866 Yuhangtang Road, Hangzhou 310058, China. [2]Fluorescence Research Group, Singapore University of Technology and Design, 8 Somapah Road, Singapore 487372, Singapore. [3]Innovation Institute for Artificial Intelligence in Medicine of Zhejiang University, 291 Fucheng Road, Hangzhou 310020, China. [4]Innovation Center in Zhejiang University, State Key Laboratory of Component-Based Chinese Medicine, Tianjin University of Traditional Chinese Medicine, Tianjin 301617, China. [5]Future Health Laboratory, Innovation Center of Yangtze River Delta, Zhejiang University, Jiashan 314100, China. [6]These authors contributed equally: Xuefeng Jiang, Min Li. ✉e-mail: zjuwangyi@zju.edu.cn; lixin81@zju.edu.cn

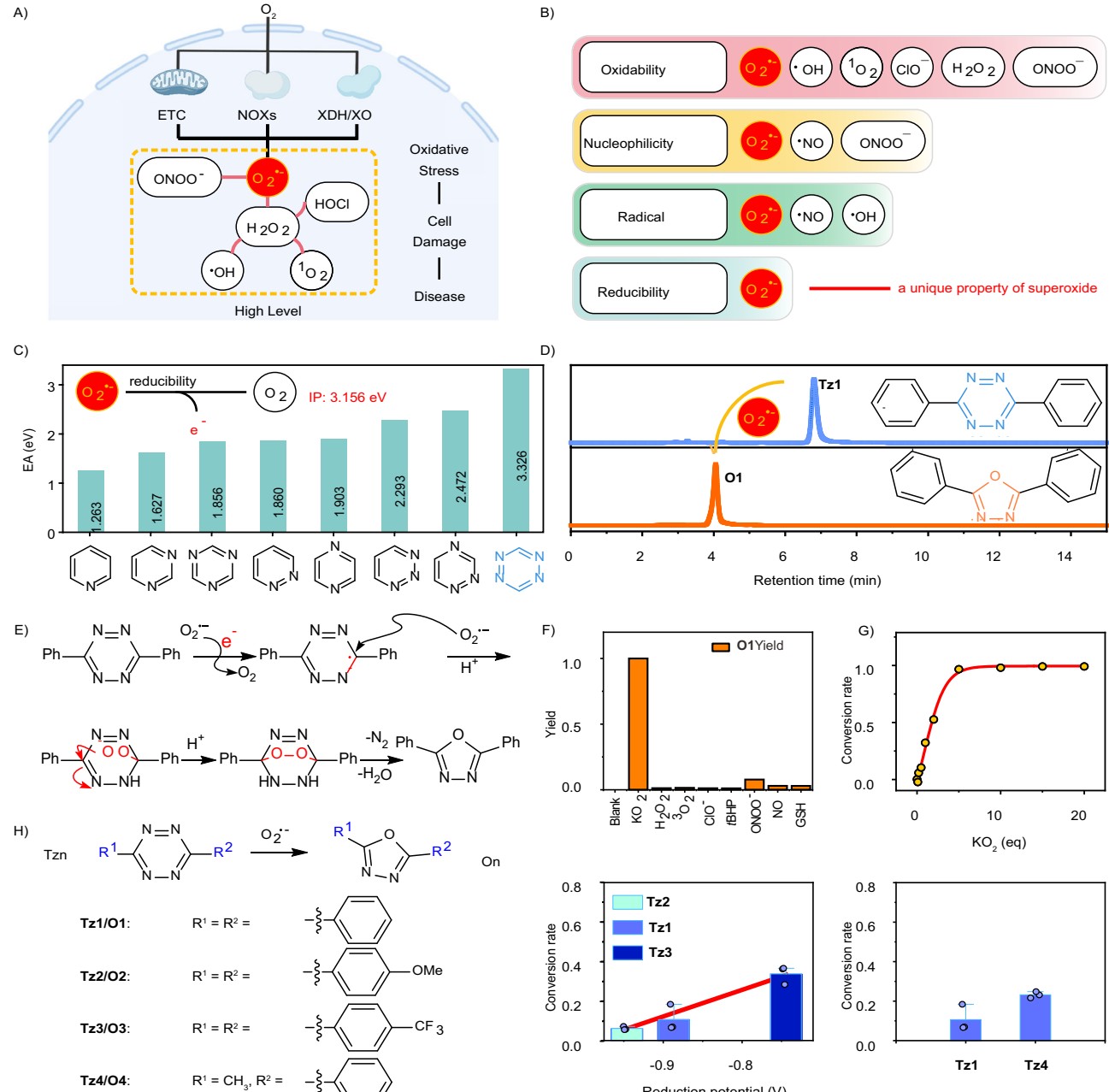

**Fig. 1 | Design of tetrazine-based probes for sensing superoxide. A** Superoxide as a primary ROS in cells. This figure was created with BioRender.com. **B** Reactivity of various ROS. **C** Nitrogen substitution of benzene ring up-shifts its electron affinity (EA); the inset shows the ionization potential (IP) of the superoxide radical anion. **D** HPLC traces of **Tz1** before and after the treatment of superoxide. **E** Proposed reaction mechanism between 1,2,4,5-tetrazine and superoxide to yield 2,5-diphenyl-l,3,4-oxadiazole (**O1**). **F** Reactivity of Tz1 when treated with various analytes. Data shown were the normalized yields of **O1**. **G** Superoxide dose-dependent conversion of **Tz1** to **O1**. **H** Superoxide transformed **Tz1**-**Tz4** to their oxadiazole counterparts, and the stereoelectronic effects on their reactivity towards superoxide. Data are presented as mean value ± SD, *n* = 3 independent experiments.

This advantage is critical, given the labile nature and the compartmentalized presence of superoxide in cells. To this end, fluorescent probes dihydroethidium (DHE) and its mitochondria-anchored analog MitoSOX have become the most popular tools for detecting superoxide in biology[24]. Blue-emissive DHE is readily oxidized by superoxide to form red emissive product(s). Recording the total intensities of red fluorescence in cells has therefore been routinely performed for assessing cellular superoxide production. Although this method has been used for about two decades, studies show that DHE can also be oxidized by other ROS to form red emissive product(s), interfering with the imaging of superoxide[25]. Further studies identified 2-hydroxyethidium as a specific product from the oxidation of DHE via

superoxide, and thereafter the high-performance liquid chromatography (HPLC) analysis of 2-hydroxyethidium was proposed to detect superoxide[26]. This assay achieved the desired specificity, but unfortunately, at the expense of losing spatiotemporal resolution. Indeed, since oxidation is a common feature of the ROS family, oxidation-based probes for superoxide generally show suboptimal specificity. To address this issue, several probes were developed based on detecting nucleophilic ROS by using sulfonylated and phosphinated probes[27–29]. Though the development of these probes improved selectivity towards superoxide over other ROS; the tendency of hydrolysis and the abundant presence of biological nucleophiles (such as glutathione) still compromise their specificity. The development of fluorescent

probes with desirable specificity for the spatiotemporal imaging of intracellular superoxide remains a daunting challenge.

Herein, we report the design and development of a family of activity-based sensing (ABS) probes for imaging superoxide in live cells with unprecedented specificity. The probes are developed based on the reductive nature of superoxide[30], and utilize the single electron transfer from superoxide to 1,2,4,5-tetrazine (Tz) as a responsive mechanism. Due to the inherent fluorescence-quenching ability of Tz[31,32], these probes show ultra-fluorogenic responses towards superoxide. By tuning probe reactivity and emission color, multiplexed imaging of cellular superoxide levels is realized with unprecedented spatial resolution. Given the robustness of the probes, we build a high-content drug screening model and identify a natural product to alleviate oxidative stress-induced injury in the pathology of ischemic heart disease. We envision that the specificity and ultra-fluorogenic response of these Tz-based probes will make them highly useful tools for tracking superoxide in a range of pathophysiological settings.

## Results

### Design of tetrazine-based probes for sensing superoxide

Activity-based sensing (ABS) conceptualized by Chang and co-workers[33,34], has proven powerful for the selective imaging of bioanalytes in live cells. ABS utilizes chemical reactivity to develop probes for the selective and sensitive detection of analytes and is especially applicable for labile species as demonstrated by the success of boronate-based probes for $H_2O_2$[35,36].

Inspired by the ABS strategy, we outlined four key requirements for developing superoxide probes. First, these probes should show desirable biocompatibility, including low cytotoxicity and high self-stability. Second, they should be highly specific to superoxide in a complex biological system, with no or minimal interference from other ROS or reactive biomolecules. Third, given the extremely transient and low concentrations of superoxide in cells, these probes should react with superoxide with ultrafast kinetics in the biological environment. Fourth, the probes should emit distinct signals, either fluorogenic or ratiometric, upon the detection of superoxide.

To design probes fulfilling these criteria, we turned our attention to the reductive property of superoxide. Superoxide may act as a moderate one-electron reducing agent[30]. It has been used to reduce benzothiazolium salts via one-electron transfer[37]. This reductive property is unique for superoxide among the ROS family and could be utilized to construct dedicated superoxide probes (Fig. 1B). In addition, 1,2,4,5-tetrazine (Tz) possesses a significant reduction potential (Fig. 1C). Studies suggest that the nitrogen substitution of the benzene ring up-shifts its reduction potential, with more nitrogen substitutions (especially the N=N bond) leading to higher reduction potentials[38]. We calculated the electron affinity (EA) of Tz and its analogs. Our results show that Tz exhibits the largest EA (3.32 eV), suggesting its strong oxidative properties (Fig. 1C). In addition, we also calculated the ionization potential (IP) of the superoxide radical anion (3.15 eV). This IP is smaller than the EA of Tz. Based on these reducing/oxidizing properties, we hypothesized that Tz may enable selective detection of superoxide by a single electron transfer (SET) mechanism. It is worth mentioning that Tz is well known for its bioorthogonal reaction with trans-cyclooctene for labeling proteins, and has demonstrated robust biocompatibility[39,40]. Moreover, Tz is a good fluorescence dark quencher for constructing fluorogenic probes[31]. We could thus develop a platter of colorful fluorogenic probes to monitor superoxide activities in live cells.

To test this hypothesis, we prepared 3,6-diphenyl-1,2,4,5-tetrazine (Tz1) and utilized it as a model compound to interrogate its reactivity with superoxide. Tz1 was treated with an excess of superoxide (administrated as $KO_2$) and the reaction was analyzed via both HPLC and liquid-chromatography-mass spectrometry (LC-MS). Our results showed the disappearance of Tz1 and the emergence of a new peak;

mass analysis showed that this peak was attributed to 2,5-diphenyl-l,3,4-oxadiazole (Fig. 1D, Supplementary Fig. 1). Three additional experiments further confirmed the molecular structure of 2,5-diphenyl-l,3,4-oxadiazole. First, the product was analyzed by high-resolution mass spectrometry (HRMS), from which we obtained consistent results (Supplementary Fig. 2). Next, 2,5-diphenyl-l,3,4-oxadiazole was prepared by a literature procedure and was used as a standard. A comparison of the standard and the product between Tz1 and superoxide gave the same retention time during HPLC analysis (Supplementary Fig. 3). Finally, after the evaporation of the reaction solvent between Tz1 and superoxide, the product residue was directly analyzed by proton nuclear magnetic resonance (NMR) spectroscopy, and the signals overlapped with those of the standard (Supplementary Fig. 4). These results collectively demonstrated that superoxide reacted with Tz1 with high efficiency to yield oxadiazole as the dominant product. We have proposed a tandem pathway initiated by one electron transfer from superoxide to Tz to explain the reaction mechanism (Fig. 1E).

Given the labile nature of superoxide, we also performed experiments to confirm that the transformation from Tz1 to the oxadiazole was indeed induced by superoxide but not its descents. To this end, the reaction between Tz1 and superoxide was conducted in the presence of TEMPO or Tiron, with the former being a superoxide dismutase mimic and the latter as an electron trap[41,42]. The presence of excess TEMPO or Tiron completely blocked this transformation (Supplementary Fig. 5), leaving Tz1 intact. These results suggested that the reactions were indeed executed by superoxide. Inspired by these results, we further investigated the specificity of Tz1 towards superoxide among the ROS family and the reductive glutathione (GSH). Aliquots of Tz1 were treated with an excess of various ROS or GSH and the reactions were analyzed by HPLC. Among all tested species, only superoxide transformed Tz1 into oxadiazole, and Tz1 remained almost inert to other species (Fig. 1F, Supplementary Figs. 6, 7, Supplementary Table 1). These results demonstrated the desirable selectivity of tetrazine-based probes towards superoxide. We also investigated the dependence of the conversion rate on the dose of superoxide, and a positive correlation was observed (Fig. 1G, Supplementary Fig. 8). This relationship suggested the potency of the probe to quantify superoxide concentrations.

To explore how the electronic effects of the substituents on Tz affected its reactivity with superoxide, Tz2 and Tz3 bearing either an electron-donating or an electron-withdrawing group on the phenyl ring were prepared. Tz1-Tz3 reacted with an insufficient amount of superoxide (0.5 eq), respectively, and these reactions were monitored both by HPLC and LCMS. As expected, all compounds reacted with superoxide to yield the oxadiazole derivatives (Supplementary Figs. 9–11). However, their conversion rates varied considerably. Tz3 bearing an electron-withdrawing group showed the highest reduction potential and the best conversion rate (Supplementary Table 2, Fig. 1H, Supplementary Fig. 12).

To test how the steric effects affected this reactivity, one phenyl group in Tz1 was changed into a methyl group to reduce the steric hindrance, affording Tz4. Tz4 reacted with superoxide with a much-improved conversion rate in comparison to Tz1 (Supplementary Fig. 13, Supplementary Table 2). It should be noted that in addition to the less steric effect, the methyl group is also less electron-donating than the phenyl group in Tz1. These observations agreed with our proposed reaction mechanism (Fig. 1E) and shed light on the molecular design of the superoxide probes with tunable sensitivity.

### Development of Tz-based fluorogenic probes for imaging superoxide in live cells

After identifying Tz as an ultra-specific responsive trigger for superoxide, we set out to design fluorogenic probes for imaging superoxide in live cells by tethering Tz to various fluorophores. We

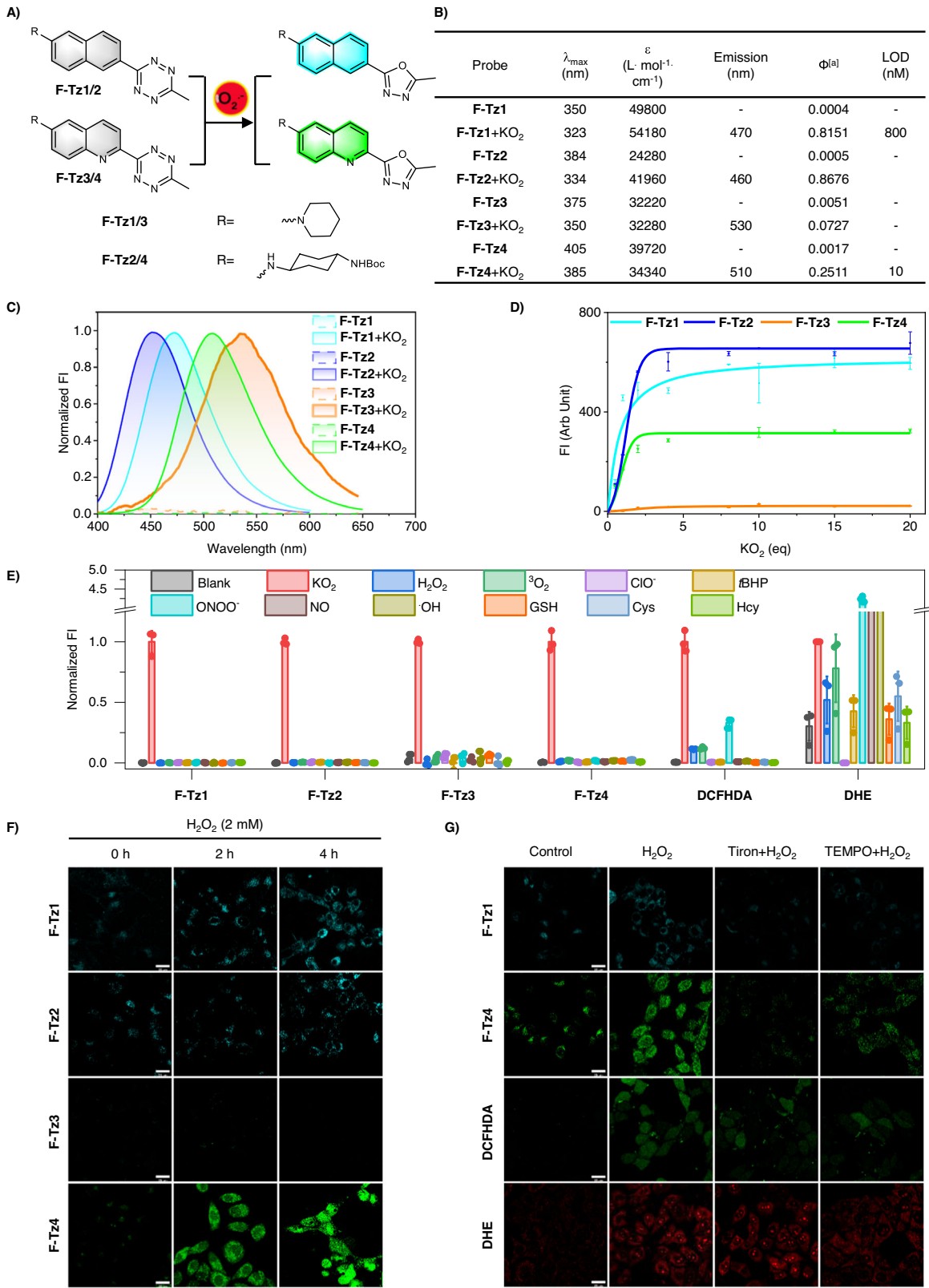

first chose naphthalene and quinoline (Fig. 2A). These two fluorophores are structurally similar but emit in different spectra ranges. The electron-withdrawing N atom at the quinolone fluorophore could render the quinoline-tetrazine probes more sensitive toward superoxide than the naphthalene-tetrazine analogs, according to our initial results (Fig. 1H). These varied sensitivities as well as their distinct emission colors could enable the multiplexed imaging of

superoxide levels in live cells. With these considerations, Tz was tethered at the C-2 position of either the naphthalene (**F-Tz1**, **F-Tz2**) or the quinoline fluorophore (**F-Tz3**, **F-Tz4**) (Fig. 2A). To tune the electron push-pull effect, amines of different electron-donating abilities were introduced at their C-6 positions. These amino groups are essential for the final naphthyl or quinolyl oxadiazoles to emit bright fluorescence, by stabilizing $\pi-\pi^*$ transitions. We facilely

**Fig. 2 | Design of Tz-based fluorogenic probes for imaging superoxide in live cells. A** Structures of the probes and their sensing mechanism. **B** Photophysical properties of the probes before and after sensing superoxide. [a]Φ: Quantum yields were determined with quinine sulfate ($\Phi_{standard} = 0.577$ in 0.1 M $H_2SO_4$) as a standard ($\lambda_{ex}$ 365 nm). **C** Normalized emission spectra of the probes (5 μM) before and after the treatment of superoxide (20 eq). The data were normalized to the maximum emission after the treatment of superoxide. **D** Plot of probe (5 μM) fluorescence intensity at their peak emission wavelengths as a function of superoxide doses. Data are presented as mean value ± SD, $n = 3$ independent experiments. **E** Fluorescence response of the probes toward various reactive analytes in comparison to DCFHDA and DHE. All probes were used at 5 μM and the reactive analytes were used at 100 μM except ONOO[−] (10 μM) and NO (20 μM). The reactions were carried out in PBS (pH 7.4, 10 mM) at ambient temperature for 30 min before

measurement. Data were the normalized emission intensities at their peak emission wavelengths. Data are presented as mean value ± SD, $n = 3$ independent experiments. **F** Confocal microscopy images of HepG2 cells pretreated with $H_2O_2$ (2 mM) for various durations and then stained with either the blue emissive **F-Tz1** or **F-Tz2**, or green emissive **F-Tz3** or **F-Tz4**. Probes were used at a final concentration of 5 μM and were incubated with the cells for 30 min before imaging. Representative images are shown from $n = 3$ independent experiments. **G** Fluorescence images of HepG2 cells stained with **F-Tz1**, **F-Tz4**, DCFHDA, or DHE (each 5 μM, 30 min). Before being stained with the probes, cells were intact (control), or pretreated with $H_2O_2$ (2 mM) for 2 h, or first pretreated with tiron (100 μM) or TEMPO (300 μM) for 1 h, and then the co-treatment of tiron (100 μM) or TEMPO (300 μM) together with $H_2O_2$ (2 mM) for 2 h. Scale bar: 25 μm. Representative images are shown from $n = 3$ independent experiments.

synthesized these probes via the procedures shown in Supplementary Information.

After obtaining these probes, we measured their optical response toward superoxide in phosphate-buffered saline (PBS) (Fig. 2B). Intact probes generally exhibited their UV-vis absorption peaks in the range of 350–405 nm; whereas superoxide treatment caused blueshifts of ~20–50 nm, due to the modification of π-conjugations (Supplementary Figs. 14–17). Since the oxadiazole scaffold generally features a shorter absorption band than the Tz moiety, these spectral changes suggested the superoxide-induced transformation of the Tz trigger into the oxadiazole moiety. These structural changes were experimentally confirmed by LCMS (Supplementary Figs. 18–21). Next, fluorescence measurements show that unreacted probes were barely emissive. However, after the treatment of superoxide, dramatic fluorescence switch-on response was observed. While **F-Tz3** and **F-Tz4** emit green fluorescence, the fluorescence of **F-Tz1** and **F-Tz2** is in the blue range (Fig. 2C, Supplementary Fig. 22). The fluorogenicity of these probes was further rationalized via quantum chemical calculations: before the reactions with superoxide, the Tz moiety generates a low-lying dark state, quenching the fluorescence; after the reactions, the Tz moiety is destroyed and the associated dark states are thus removed, activating bright fluorescence from the resulting fluorophores (Supplementary Figs. 23–26)[43].

We noted that the fluorescence intensities of all these four probes depend on the dose of superoxide (Fig. 2D, Supplementary Fig. 27). Moreover, their fluorogenic responses are highly specific towards superoxide, as other biologically relevant species posed almost no interference (Fig. 2E, Supplementary Fig. 28). This specificity is superior to the commercial probes DHE and DCFHDA. DHE is regarded as a superoxide-specific probe and DCFHDA is a nonselective ROS probe. However, parallel experiments showed that both DHE and DCFHDA can be turned on by various ROS species, exhibiting poor selectivity (Fig. 2E, Supplementary Figs. 29, 30). We noted that the tetrazine-based probes demonstrated much improved storage stability than DHE and DCFHDA (Supplementary Figs. 31–34), in addition to their desirable selectivity.

Before cell imaging, cell viability assay was performed to confirm the safety of these probes (Supplementary Fig. 35). Moreover, their oxadiazole derivatives which were prepared by treating the probes with superoxide in flasks, were also confirmed to show negligible cytotoxicity (Supplementary Fig. 35). We then evaluated their performance in detecting superoxide in live cells. To induce robust upregulation of cellular superoxide, HepG2 cells were stimulated with $H_2O_2$ (2 mM) for various durations. We first confirmed that $H_2O_2$ even at this high concentration couldn't react with the Tz-based probes by both fluorescence spectra (Supplementary Fig. 36) and LC-MS analysis (Supplementary Fig. 37). Then the cells were stained with the probes for 30 min and imaged under confocal microscopy. All probes except **F-Tz3** were able to image superoxide in HepG2 cells (Fig. 2F), and their fluorescence intensity positively correlated with the $H_2O_2$-incubation time (Supplementary Fig. 38). The failure of **F-Tz3** in the cell imaging

assay was probably due to the low brightness of the reaction product or the low retainability in cells. Based on the brightness data, the highly emissive blue probe **F-Tz1** and the green probe **F-Tz4** were used in the subsequent studies.

Live cells undergoing oxidative stress inevitably express a variety of ROS. Specific probes for a single ROS are thus highly sought-after. To confirm the specificity of **F-Tz1** and **F-Tz4** towards superoxide in live cells, we used the method of superoxide-specific scavenging. Tiron and TEMPO are two reagents that efficiently scavenge superoxide[41,42]. These two reagents were then used in the $H_2O_2$-treated cells to decrease cellular superoxide levels. Accordingly, both Tiron and TEMPO worked efficiently to inhibit the fluorogenic response of probes **F-Tz1** and **F-Tz4** in such live HepG2 cells. In contrast, they only partially inhibited the fluorescent response of DHE and DCFHDA (Fig. 2G and Supplementary Fig. 39). These results are in accordance with the better selectivity of the tetrazine-based probes towards superoxide than DCFHDA and DHE (Fig. 2E) and highlight the unprecedented specificity of the Tz-based probes **F-Tz1** and **F-Tz4** for imaging cellular superoxide.

In addition to monitoring the elevated superoxide levels in live cells with exogenous oxidant treatment, probe **F-Tz4** was also confirmed to be capable of imaging endogenous superoxide during toxin-induced oxidative stress or physiological redox signaling. Paraquat is known to upregulate cellular superoxide[44]. HepG2 cells pretreated with paraquat (0.5, 1, or 3 mM) for 24 h were then stained with **F-Tz4**. Confocal imaging results showed that the cellular probe fluorescence intensity was positively correlated to paraquat doses that the cells were exposed (Supplementary Fig. 40). Next, a moderate upregulation of cellular superoxide levels was induced by treating A549 cells expressing the epidermal growth factor receptor (EGFR) with epidermal growth factor (EGF), a process known to activate NOX activity. A549 cells were first stimulated with EGF (0.5 μg/mL) and then stained with **F-Tz4**. Compared with the vehicle control, the EGF group showed significantly increased cellular probe fluorescence, and this increase could be compromised by the pretreatment of cells either with DPI (5 μM) which is an unspecific NOX inhibitor, or by VAS2870 which inhibits NOX with fair specificity (Supplementary Fig. 41)[45]. These results suggest that **F-Tz4** is sensitive enough to image endogenous superoxide.

## Multiplexed imaging of cellular superoxide with high spatial resolution

Inspired by the different emission colors of **F-Tz1** and **F-Tz4** and their considerably different limit of detection (LOD) towards superoxide (Fig. 2B, Supplementary Figs. 42, 43), we evaluated the possibility to quantify cellular superoxide levels. For this purpose, HepG2 cells were pretreated with various concentrations of $H_2O_2$ for 2 h to induce varying levels of cellular superoxide. The feasibility of this model was first confirmed by feeding the cells with DHE and then quantifying cellular 2-hydroxyethidium levels with LCMS analysis which is a known superoxide-specific product. While 2-hydroxyethidium was almost

absent in the starting reagent, its levels increased as the cells were stimulated with increasing doses of $H_2O_2$ (Supplementary Fig. 44). This result implied that adjusting $H_2O_2$ doses or duration time should be feasible to tune cellular superoxide levels. After establishing this model, the cells pretreated with various doses of $H_2O_2$ were then stained with either **F-Tz1** or **F-Tz4**. As shown by the confocal imaging results (Fig. 3A, C, and Supplementary Figs. 45, 46 for brightfield images), a significant fluorogenic response from **F-Tz1** was observed only in the high $H_2O_2$-dosage group, while **F-Tz4** turned on bright fluorescence even in the low $H_2O_2$-dosage group.

Similar results were observed when cells were treated with the same dose of $H_2O_2$ (2 mM) for different durations (Fig. 3B, D, and Supplementary Figs. 45, 46 for brightfield images). These results were in good agreement with the different sensitivity of the two probes towards superoxide (Fig. 2B) and confirmed the possibility of detecting cellular superoxide levels with the combinational applications of various Tz-based probes.

Interestingly, the subcellular distribution of probe **F-Tz4** alone could also be used to indicate oxidative stress levels. When HepG2 cells were treated with $H_2O_2$ of low dose (0.5 mM) for 2 h and then stained with **F-Tz4**, fluorescence was observed only in the cytoplasm compartment and colocalized well with the signal from Mito Tracker (Fig. 3E). When HepG2 cells were stimulated with $H_2O_2$ of high dose (2 mM) for 2 h followed by similar staining procedures, fluorescence could also be observed in the nucleus, as shown by the co-localized signals from DHE. Herein DHE was used as a nuclear stain because oxidized DHE tends to accumulate in the nucleus via intercalating into DNA[46]. Meanwhile, the colocalization of **F-Tz4** with Mito Tracker greatly decreased (Fig. 3F). We assumed that under low levels of oxidative stress, mitochondria remained the dominant place for superoxide production. Therefore, the fluorescence from **F-Tz4** colocalized well with that of Mito Tracker. However, when cells were challenged with a high degree of oxidative stress, a variety of oxidases were activated including those localized in the nucleus. The burst of superoxide production by these oxidases then changed the distribution of fluorescence signals from **F-Tz4**. The emergence of fluorescence signals in the nucleus is thus associated with a high degree of cellular oxidative stress.

### Extension of the strategy to other fluorophores

Encouraged by the desirable specificity of Tz towards superoxide detection, its excellent fluorescence-quenching ability, and module design with various fluorophores, we then extended this probe strategy to other fluorophores with emission wavelengths ranging from bluish violet to red (Fig. 4A). Coumarine, naphthalimides, rhodamine, and acridine orange have been frequently used in cell imaging experiments, owing to their good photophysical properties. After being tethered with Tz (Fig. 4A), these fluorophores all demonstrated negligible fluorescence. After the treatment of superoxide, an ultra-fluorogenic response was observed (Fig. 4B, C), with fluorescence turn-on ratios up to -1400 times. Quantum chemical calculations show that this ultrafluorogenicity is related to the reaction of the Tz moiety (Supplementary Figs. 47–51)[43,47].

Our subsequent experiments confirmed that this fluorescence turn-on response is highly specific towards superoxide (Fig. 4B), resulting in the conversion from Tz to oxadiazole (Supplementary Fig. 52). The ability of these probes to image superoxide in live cells was also demonstrated (Fig. 4D, Supplementary Fig. 53).

### Tz-based probes facilitated the high-content screening for superoxide modulators to suppress myocardial infarction-induced injury

After verifying the desirable specificity of Tz-based probes for imaging superoxide in live cells, we then tested if these probes could be utilized to construct a high-content screening model for superoxide modulators, by monitoring the fluorescence intensity changes. It should be noted that with the clinical failure of most low-molecule weight antioxidants that stoichiometrically scavenge ROS un-selectively, and with the more recognized notion that ROS are also important messengers in redox signaling, it emerges as a more plausible way to search for redox modulators to conteract oxidative stress[4]. We therefore ascertain that a high-content screening model should be highly relevant for such a purpose.

Myocardial infarction is a deadly medical condition with high incidence. A major cause of myocardial infarction is oxidative stress[48]. To confirm the etiological role of superoxide in myocardial infarction, the accumulation of superoxide during myocardial ischemia/reperfusion (I/R) injury was ex vivo imaged with our probes. **F-Tz4** (2 mg/kg) was intra-cardic injected into the left ventricle of mice suffering from myocardial I/R injury, and cardiac sections were harvested and observed via a spinning disk confocal microscope. An approximately 2.81-fold increase of fluorescent intensity was found in mice with myocardial I/R injury compared with that of a sham-operated group (Fig. 5A). These results suggest the participation of superoxide in myocardial infarction.

After confirming the implication of superoxide in the etiology of myocardial infarction, we then set out to formulate a high-content screening model for superoxide modulators. It is important to note that many antioxidants exhibited therapeutic potential in preclinical studies but hardly achieved success in clinical trials. This discrepancy is presumably due to the ineffective scavenging of ROS, or their delayed administration at late reperfusion[4,48]. However, this shouldn't diminish the potential of antioxidants as cardioprotective agents. In this context, the search for effective superoxide modulators remains crucial for the development of cardioprotective agents. We chose to use H9C2 mice cardiomyocytes for the high-content screen. Although H9C2 cells are non-contractile and lack the ROS-generating property of normal cardiac contractility[49–51], however, considering that the elevated ROS level during oxidative stress injury was much higher than that in physiological cardiac contractility, and the availability of cultured myocardial cells, we think a high-content screening model employing H9C2 cells should be acceptable; and further validation experiments can be performed in primary neonatal rat cardiomyocytes.

First, cells were subjected to *tert*-butyl hydroperoxide (*t*BHP) treatment to trigger superoxide accumulation. The magnitude of this accumulation was monitored by either the classical dye MitoSox or our probe **F-Tz4**. *t*BHP-induced superoxide generation can be represented by the elevated fluorogenic response of both probes in a dose-dependent manner (Supplementary Fig. 54). Noteworthy, **F-Tz4** was more sensitive than MitoSox by exhibiting higher variations in fluorogenic responses, suggesting the super sensitivity of **F-Tz4**. Through an orthogonal experiment (Supplementary Fig. 55), we confirmed the optimized concentrations of *t*BHP and **F-Tz4** were 150 μM and 1 μM, respectively, for this assay.

The high-content screening was then carried out over a natural product library containing 223 compounds using the ImageXpress Micro Confocal system (Molecular Devices). After incubated with each of these compounds (25 μM) for 24 h, H9C2 cells were treated with 150 μM *t*BHP for 2 h to trigger superoxide over-generation, and subsequently stained with **F-Tz4** (Fig. 5B). The inhibitory rate of each candidate was calculated as $[(I_M - I_A)/I_M] \times 100\%$, where $I_M$ was the signal of cells with only *t*BHP treatment, and $I_A$ was the signal of cells with pre-incubation of various compounds. The majority of compounds exhibited few or no effects on superoxide accumulation (Supplementary Data 1). However, four compounds showed over 50% inhibitory activity at 25 μM (Supplementary Fig. 56). Noteworthy, three of the four hits, i.e., methylcobalamin, 7,8-dihydroxyflavone, and sinomenine have been previously reported to inhibit oxidative stress[52–54], suggesting the reliability of this model. The fourth hit, coprostanone (5α-Cholestan-3-one, 5αCh3), is a microbial metabolite

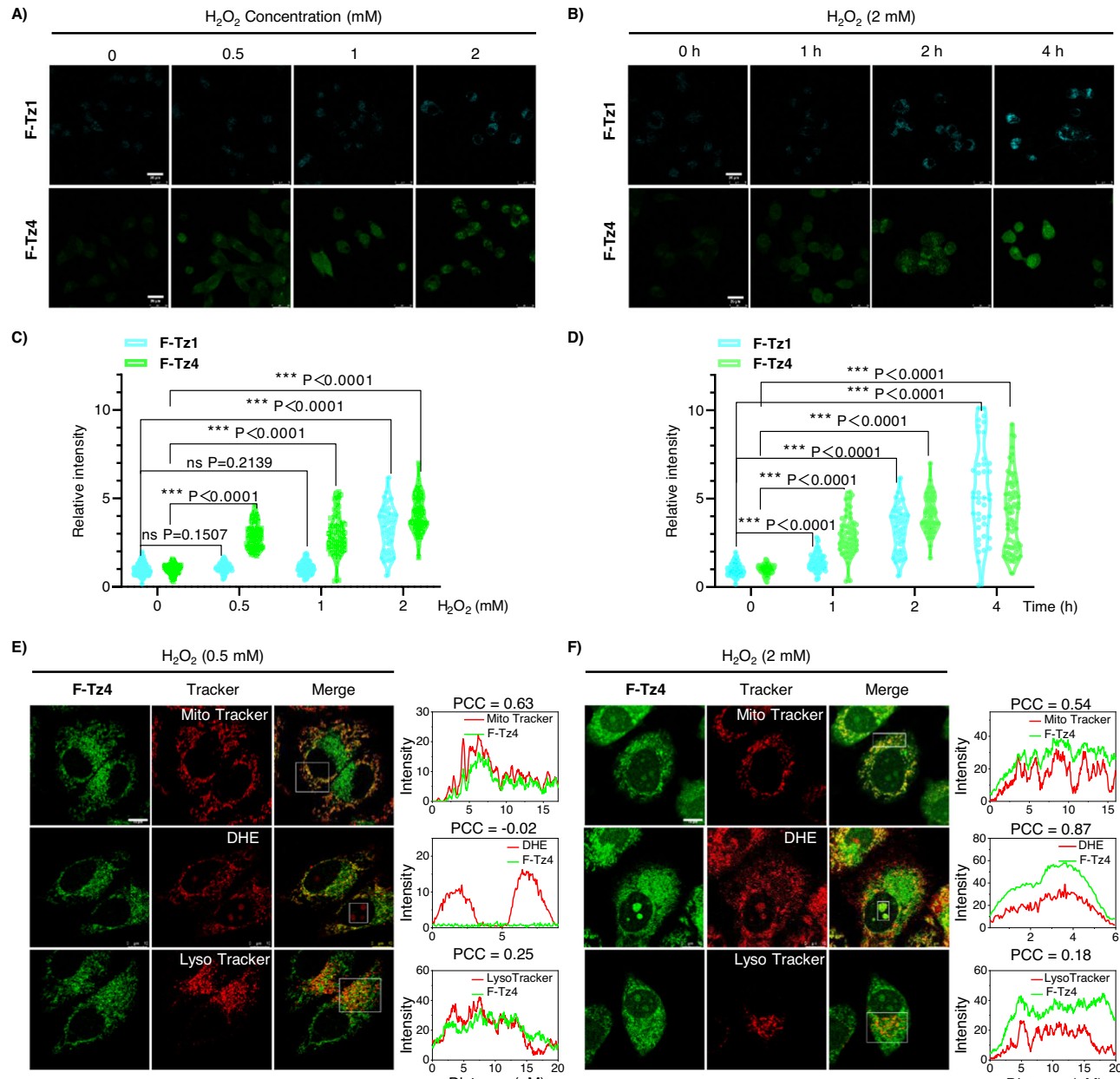

**Fig. 3 | Multiplexed imaging of cellular superoxide levels with probes F-Tz1 and F-Tz4. A, B** Fluorescence images of HepG2 cells stained with either **F-Tz1** or **F-Tz4** (each 5 μM, 30 min). Cells were pretreated with various doses of $H_2O_2$ for 2 h (**A**) or 2 mM $H_2O_2$ for various time (**B**). Scale bar: 25 μm. **C, D** The statistically quantified data on the cellular fluorescence intensity in (**A**) and (**B**). The data were normalized to the control group, and *P* values were analyzed by two-tailed unpaired t-test, 95% Confidence interval. **F-Tz1:** *n* = 97 cells for 0 mM $H_2O_2$ 0 h, *n* = 45 cells for 0.5 mM $H_2O_2$ 2 h, *n* = 66 cells for 1 mM $H_2O_2$ 2 h, *n* = 49 cells for 2 mM $H_2O_2$ 2 h, *n* = 56 cells for 2 mM $H_2O_2$ 1 h, *n* = 41 cells for 2 mM $H_2O_2$ 4 h; **F-Tz4:** *n* = 116 cells for 0 mM $H_2O_2$

0 h, *n* = 78 cells for 0.5 mM $H_2O_2$ 2 h, *n* = 68 cells for 1 mM $H_2O_2$ 2 h, *n* = 77 cells for 2 mM $H_2O_2$ 2 h, *n* = 68 cells for 2 mM $H_2O_2$ 1 h, *n* = 74 cells for 2 mM $H_2O_2$ 4 h. All cell numbers are over three biologically independent experiments. **E, F** Fluorescence images of HepG2 cells co-stained with **F-Tz4** (5 μM, 30 min) and various organelle markers (50 nM Mito Tracker Red CMXRos, 50 nM Lyso Tracker Red, or 5 μM DHE for nuclei, 15 min). Cells were pretreated either with a low (0.5 mM) or high (2 mM) dose of $H_2O_2$ for 2 h before being stained with the probes. The right panel showed the intensity profile and PCC (Pearson's correlation coefficient) along the white rectangle highlighted in the left panel. Scale bars, 10 μm.

of cholesterol (Fig. 6A). Coprostanone has not been reported for its redox modulating activity[55].

To validate this activity of coprostanone, we tracked time-dependent overproduction of superoxide in H9C2 cells induced by 150 μM *t*BHP, with the cells pretreated (24 h) with coprostanone from 10 to 50 μM. Fluorescence images were recorded every 15 min by the high-content screening system. As shown in Fig. 6B, C and Supplementary Fig. 57, coprostanone inhibited the increase of probe fluorescence intensity in a dose-dependent manner, demonstrating its potency in preventing superoxide overproduction. Furthermore, we

confirmed the effect of coprostanone to inhibit *t*BHP-induced super-oxide overproduction in primary mice cardiomyocytes. While 30 μM *t*BHP treatment significantly upregulated cellular probe fluorescence, pretreating the primary mice cardiomyocytes with coprostanone dose-dependently compromised this effect (Supplementary Fig. 58). This result suggested that the screening results obtained in the H9C2 cell line should be translated to primary mice cardiomyocytes.

Inspired by this result, we moved on to test the cardioprotective effects of coprostanone in a mice model with myocardial I/R injury. Mice were gavage administrated with coprostanone (50 mg/kg/d or

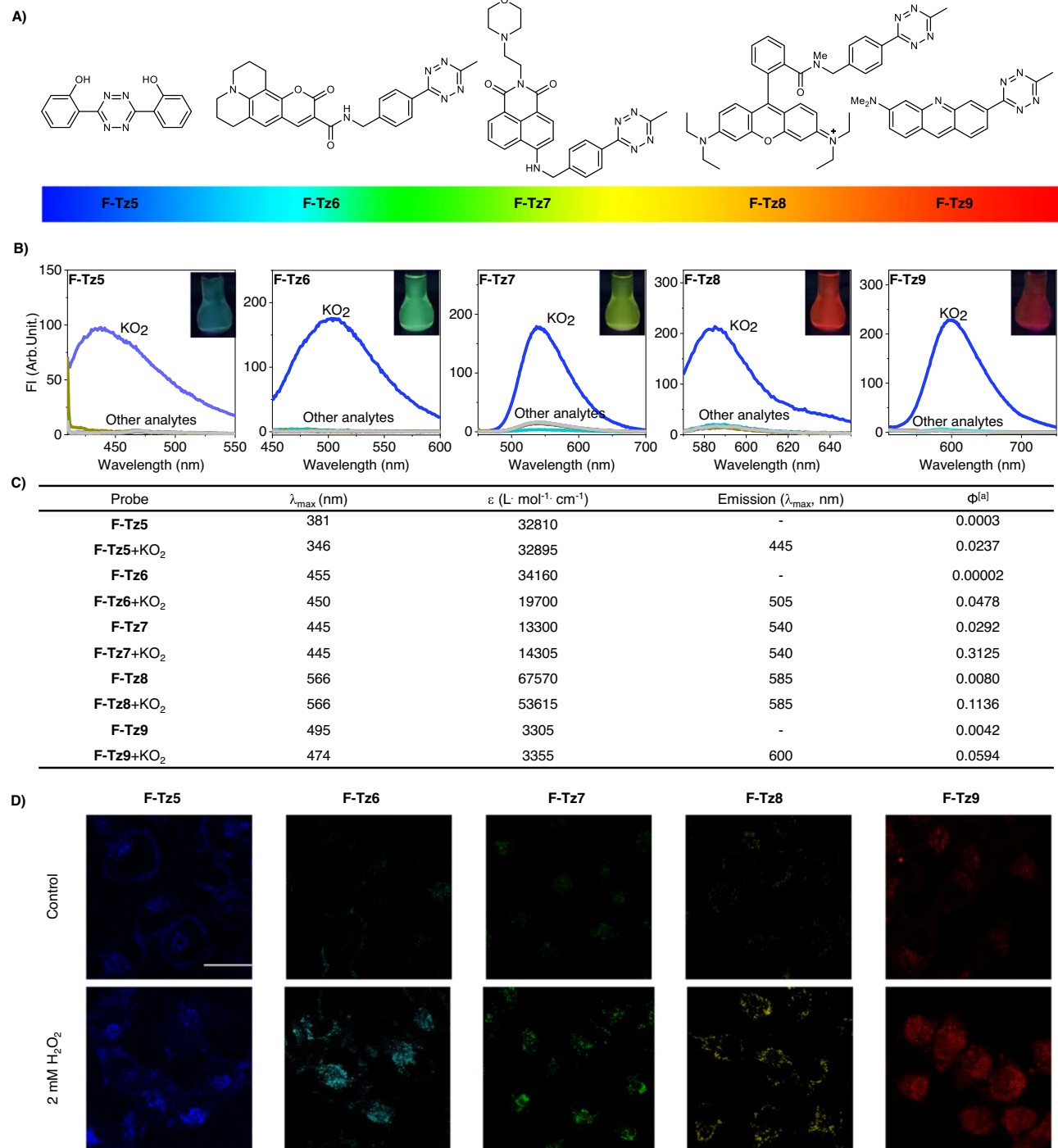

**Fig. 4 | Extension of Tz to other fluorophores to tune probe emission spectra.**
**A** Structures of the Tz-tethered probes. **B** Fluorescence spectra of the probes
before or after the treatment of various analytes. **C** Photophysical properties of
the probes before or after the treatment of superoxide. [a]Φ: Quantum yields of **F-Tz5/6** were determined at an excitation wavelength of 365 nm and using quinine
sulfate ($\Phi_{standard}$ = 0.577 in 0.1 M $H_2SO_4$) as a standard. Quantum yields of **F-Tz7**
were determined at an excitation wavelength of 460 nm and using fluorescein
($\Phi_{standard}$ = 0.95 in 0.1 M NaOH) as a standard. Quantum yields of **F-Tz8/9** were
determined at an excitation wavelength of 470 nm and using fluorescein ($\Phi_{standard}$
= 0.95 in 0.1 M NaOH) as a standard. **D** Imaging superoxide in live cells with these
probes. Scale bar: 25 μm. Representative images are shown from $n$ = 3 independent experiments.

100 mg/kg/d) for 3 days, and then were subjected to surgical I/R injury.
Compared with I/R mice, coprostanone pretreatment significantly
reduced TTC-stained infarct size (Fig. 6D, E). To test the effects of
coprostanone on hemodynamics in mice of sham operation or I/R
injury, mice were randomly divided into four groups, including sham,
sham + 5αCh3 (100 mg/kg), I/R, and I/R + 5αCh3 (100 mg/kg) groups.
After 3 days of pretreatment with 5αCh3 or vehicle, I/R and sham

operations were performed. After 45 min ischemia followed by 24 h
reperfusion, each anesthetized mouse was micro-cannulated with a
1.4 F microcatheter. Parameters, including left ventricular systolic
pressure (LVSP), left ventricular end-diastolic pressure (LVEDP) and
the maximum and minimum rates of left ventricular pressure changes
(dp/dt max and dp/dt min), were automatically recorded. Compared
with the sham-operated group, the abnormal values of LVSP and dp/dt

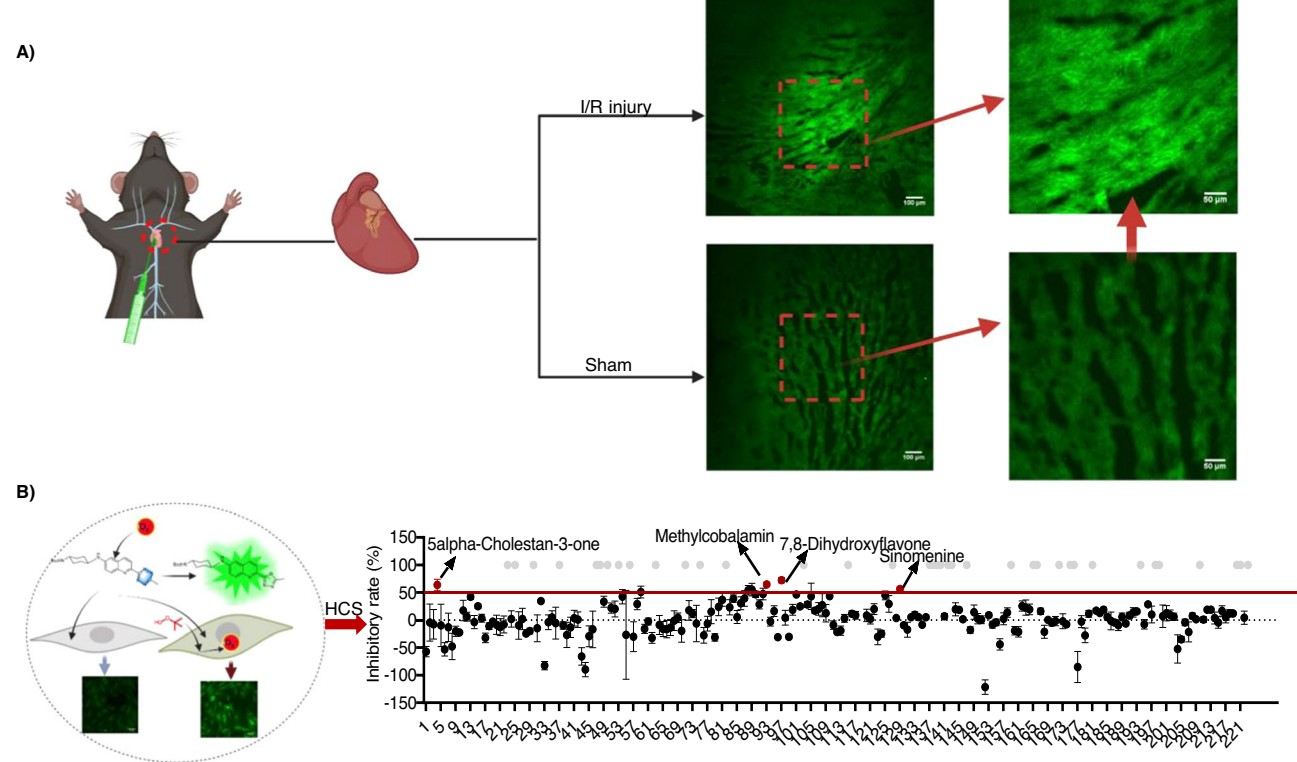

**Fig. 5 | High-content screening for superoxide modulators from natural products by F-Tz4 staining. A** Ex vivo **F-Tz4** imaging of superoxide production during myocardial I/R injury. $n = 4$ mice. **B** The illustration of cell imaging schematic and high content screening workflow (created with BioRender.com). For the screening results, the inhibition rate greater than 50% was set as the screening standard. Each compound was repeated three times.

max were observed in mice subjected to I/R injury. The pretreatment of 5αCh3 effectively improved these parameters in mice subjected to I/R injury; while it caused no significant effects on these parameters in the sham group (Supplementary Fig. 59). Apart from eliminating the side effects of 5αCh3 on hemodynamics of sham-operated mice, these data highlighted the beneficial effects of 5αCh3 against the impaired hemodynamic parameters of I/R mice.

To further investigate the potential pharmacological mechanism of coprostanone, the expression of oxidative stress-related proteins in myocardial tissues was analyzed by Western blotting. We found that the NRF2-mediated transcriptional antioxidant program, including its downstream factors HO-1 and SOD2, was significantly regulated by coprostanone treatment (Fig. 6F). Moreover, an immunofluorescence assay was carried out, which showed that the heart tissues from mice pretreated with coprostanone and then subjected to I/R injury expressed HO-1 and SOD2 at higher levels than the vehicle group (Supplementary Fig. 60). While the exact mechanism by which coprostanone induces cardioprotection remains to be explored, its effect to inhibit superoxide overload has been established, and the enhancement of the NRF2-HO-1/SOD2 signaling pathway may play a role in this effect.

## Discussion

Redox regulation is key to maintain cell homeostasis. Imbalanced redox contributes to the occurrence and development of various aging-related diseases. Although antioxidants are reported to reduce oxidative stress and increase healthy longevity[56], several antioxidants showed detrimental effects in clinical trials[57]. These controversial results suggest the complexity of redox signaling and advocate the necessity of in-depth studies of various redox species with precision[58]. It is therefore urgent to develop reliable assays to monitor each of these redox species with high selectivity.

Superoxide is the major initial form of ROS. It is readily converted to $H_2O_2$ via superoxide dismutase enzyme[9], which is further transformed into •OH through Fenton reaction or into HClO by myeloperoxidases[10,11]. Superoxide can also interact with nitric oxide to form peroxynitrite[12]. Evidence suggests that approximately 0.2–2% $O_2$ consumed by mitochondria is converted into superoxide under normal physiological conditions, fueling the generation of $H_2O_2$ and other ROS for redox signaling[59]. Its excessive generation or inefficient dismutation ignites oxidative stress, causing the pathogenesis of many diseases[13–17]. Therefore, the spatiotemporal-resolved detection of superoxide is of primary importance.

There has been continuous efforts to develop assays for detecting superoxide[18], as summarized in Supplementary Data 2. However, no current method is without limitation. Assays such as the DHE-HPLC assay[26], or the EPR assay[19], fulfill the selectivity requirement but lack spatiotemporal resolution. Assays such as DHE/MitoSox imaging can monitor superoxide in live cells, but the selectivity is compromised due to their cross-reactivity towards various ROS[24].

Fluorescent imaging is a desirable modality to realize both the selective and spatiotemporal-resolved detection of superoxide in live cells. We thus set out to develop selective superoxide-responsive fluorescent probes. Judiciously comparing the chemical reactivity of various ROS revealed that superoxide possesses unique reducibility[1]. It is capable of transferring one electron to other compounds with suitable reduction potentials. This reactivity is opposite to other ROS which tends to extract one electron from other compounds. Based on this distinct reactivity and with the knowledge that multiple nitrogen-substituted benzenes have a strong tendency to be reduced, we hypothesized that 1,2,4,5-tetrazine could act as a selective superoxide-responsive trigger. We calculated the electron affinity of 1,2,4,5-tetrazine to be 3.326 eV, which is considerably greater than the ionization potential of the superoxide radical anion (3.156 eV), suggesting the

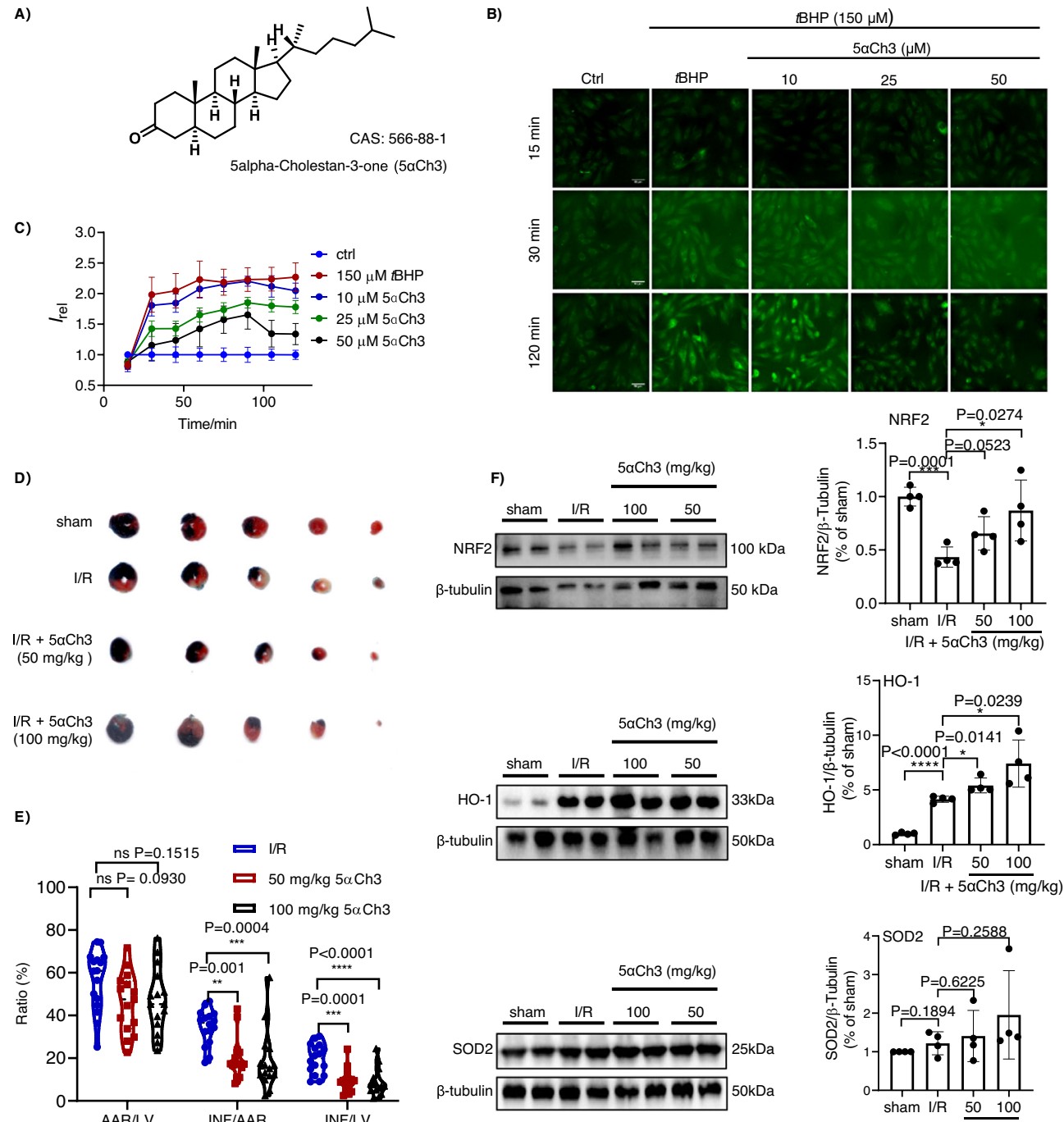

**Fig. 6 | Verifying the effect of coprostanone to inhibit superoxide over-production and to ameliorate myocardial I/R injury in mice. A** The molecular structure of coprostanone. **B** Representative images of H9C2 cells stimulated with *t*BHP and stained with **F-Tz4** (1 μM). Cells were intact (control), treated with *t*BHP (150 μM), or pretreated with coprostanone (10, 25, 50 μM) for 24 h and then treated with *t*BHP (150 μM). All cells were co-treated with probe **F-Tz4** at the time *t*BHP was adminstrated. Scale bar: 50 μm. **C** The statistically quantified data on the cellular fluorescence intensity of cells pretreated with coprostanone (24 h) of different doses, and then treated with *t*BHP and **F-Tz4**. The data were the mean ± SD and were normalized to the control group. **D** Representative photographs of TTC staining of myocardial infarction in mice subjected to myocardial I/R injury with coprostanone pretreatment for 3 days before ischemia. All the mice were anesthetized with tribromoethanol at a dose of 150 mg/kg before surgery. Mice

myocardial ischemia and reperfusion injury were induced by the occlusion of the left anterior descending coronary artery for 45 min followed by 24 h reperfusion. The sham-operated group underwent the same procedure without ligation of the left anterior descending coronary artery. **E** TTC staining ratios of myocardial infarction in mice subjected to myocardial I/R injury (*n* = 16 mice for I/R group, 15 for 50 mg/kg and 100 mg/kg 5αCh3 group). Data were expressed as mean ± SD and analyzed with one-way ANOVA follwed by Dunnett's multiple comparisons test. \*\**P* < 0.01 vs. I/R group, \*\*\**P* < 0.001 vs. I/R group, \*\*\*\**P* < 0.0001 vs. I/R group; TTC, 2,3,5-triphenyltetrazolium chloride. **F** Representative Western Blot of NRF2, HO-1, and SOD2 in cardiac tissue lysates, *n* = 4 mice per group. All data were expressed as means ± SD and were analyzed using unpaired two-tailed t-test, \**P* < 0.05, \*\**P* < 0.01, \*\*\**P* < 0.001 versus I/R group.

potential of 1,2,4,5-tetrazine to accept one electron transferred from superoxide radical anion for its detection.

By first preparing a model compound **Tz1** and studying its reactivity towards superoxide, we excitedly confirmed the selective reaction between 1,2,4,5-tetrazine and superoxide to produce l,3,4-oxadiazole, and confirmed that this tetrazine was basically inert towards other ROS species (Fig. 1F). Importantly, the conversion rate was observed positively correlating to superoxide doses (Fig. 1G), suggesting the potential of this reaction to make a quantitative detection. Furthermore, we revealed that tetrazines with larger reduction potentials and less steric hindrance should react with superoxide more efficiently (Fig. 1H). This simple structure-sensitivity relationship should inspire the design of probes with different sensitivity towards superoxide.

1,2,4,5-Tetrazines usually have sufficient biocompatibility and stability, as evidenced by their wide application as bioorthogonal compounds for labeling proteins[40]. Due to their inherent and effective fluorescence quenching ability[31], their translation into superoxide-responsive fluorogenic probes is straightforward. We have confirmed that the tetrazine moiety can be simply tethered to a platter of fluorophores from bluish violet to red (Fig. 4A), converting them into fluorogenic superoxide probes, which demonstrates the generalizability and modularity of our design strategy. 1,2,4,5-Tetrazine-based probes fluorogenically responded to superoxide in a dose-dependent way, and this response could only be triggered on by superoxide but not other ROS species. Specifically, we have confirmed the superior selectivity of these probes to commercial probes DHE or DCFHDA by both solution-based and cell-imaging based assays (Fig. 2E, G). In addition, the probes also favor a high degree of sensitivity towards endogenous superoxide. When cells were first stimulated with paraquat to induce the upregulation of superoxide[44], and then stained with **F-Tz4**, paraquat-dose dependent increase of cellular probe fluorescence was observed (Supplementary Fig. 40). Moreover, when cells highly expressing EGFR were stimulated with EGF to induce a fast redox signaling, the burst of cellular superoxide could also be monitored by **F-Tz4** (Supplementary Fig. 41). These results suggest the high sensitivity of the probe. Moreover, we observed that NOX inhibitor DPI and VAS2870 could compromise cellular **F-Tz4** fluorescence induced by EGF, further suggesting the high selectivity of this probe towards superoxide.

Interestingly, Tz-based fluorogenic probes with different sensitivity could be used in combination to discriminate cellular oxidative states, as shown by probe **F-Tz4** and **F-Tz1**, with the former being green-emissive and highly sensitive (LOD 10 nM), while the latter blue-emissive and less sensitive (LOD 800 nM). Low levels of cellular oxidative stress resulted in green fluorescence of **F-Tz4**; while high levels of oxidative stress were visible by both channels (Fig. 3A, B). Further, **F-Tz4** itself was observed to be able to detect cellular oxidative states by the organelle distribution of its fluorescence, with the emergence of nuclear fluorescence suggesting the worsening of oxidative stress (Fig. 3F).

Given the high selectivity and sensitivity of **F-Tz4** in cell imaging experiments, it was further used to image the aberrant superoxide generation in the pathology of mice myocardial I/R injury, and thereafter to construct a high-content screening model for superoxide modulators. By employing the fluorescence output to monitor superoxide concentrations and infer the effect of candidate compounds in inhibiting oxidative stress, this screening model enabled the identification of coprostanone as a promising compound for preventing superoxide overload and ameliorating myocardial I/R injury at least in part by inducing native antioxidant enzymes. Specifically, with the failure of many ROS scavengers in clinical trials, the search for inducers of endogenous antioxidant enzymes is emerging as a more promising strategy for antioxidant development. The results herein suggests that this superoxide-specific-probe-facilitated high-content

screening model should be especially appealing for finding such inducers from either natural product libraries or traditional Chinese medicines.

These results underscore the versatility and utilities of Tz-based superoxide probes. We envision that the unprecedented specificity and spatial/temporal resolution of these Tz-based probes would warrant numerous applications to a wide range of pathological conditions. Meanwhile, it should also be noted that although we carried out a simple probe structure-activity relationship study in this work, this is far from being sufficient to obtain the most sensitive probes. This is especially true for those probes emitting in the blue-violet or red region. As shown by the data in Fig. 4C, the quantum yields of most probes after reacting with superoxide remained low, and further work to improve their sensitivity is needed.

# Methods

## Ethical statement
All animal studies were approved by the Ethics Committee for Animal Experiments of Zhejiang University in China, and performed in accordance with the Guidelines for the Care and Use of Laboratory Animals of Zhejiang University. The approved protocol number is ZJU20220096. All mice used in this study were male adult C57BL/6 mice. Male mice were used because the basic studies demonstrated estrogen treatment prevents apoptosis and necrosis of cardiac and endothelial cells causing unexplained impact on drug efficacy studies.

## Cell culture
A549, H9C2, and HepG2 cells were kindly provided by Stem cell bank, Chinese Academy of Sciences. All of them were STR-proved by stem Cell Bank, Chinese Academy of Sciences. HepG2 and A549 cells were cultured in high glucose Dulbecco's Modified Eagle Medium (DMEM, Gibico) supplemented with 10% fetal bovine serum (FBS, PAN) with 1% antibiotics (100 U/mL penicillin and 100 μg/mL streptomycin) at 37 °C and 5% $CO_2$. Cells were carefully harvested and split when they reached 80% confluence to maintain exponential growth. H9C2 cells were cultured in high glucose Dulbecco's modified Eagle medium (DMEM, Corning) supplemented with 10% fetal bovine serum (FBS, Corning) with 1% antibiotics (100 U/mL penicillin and 100 μg/mL streptomycin, Giboco) at 37 °C and 5% $CO_2$. Cells were carefully harvested and split when they reached 80% confluence to maintain exponential growth.

The cardiac myocytes were prepared from SD rats born at 0-3 d (P0-P3) (provided by Zhejiang Academy of Medical Sciences) using the Neonatal Heart Dissociation Kit (MACS, Germany) according to the manufacturer's protocol. Cardiomyocytes were plated in 96-well plates in a plating medium containing 10% serum. After 24 h of plating, the medium was then replaced with a serum-free maintenance medium and incubated for another 24 h before being used for further study.

## MTT assay
Cells of the logarithmic growth phase were taken and inoculated in 96-well culture plates with edge-well PBS replenishment and incubated in a 37 °C incubator with 5% $CO_2$. After cell apposition, the cells were administrated with complete medium containing the tested compounds and incubated for 24 h. Then replace the liquid in all wells with DMEM medium containing 0.5 mg/mL MTT. Put back into the incubator at 37 °C, and then replace the liquid in the wells with 100 μL DMSO after 4 h. After shaking for 10 min at 37 °C, the absorbance of 580 nm was measured by an microplate reader TECAN infinite M1000 Multi-function microplate reader. The absorbance of each well was compared with the absorbance of normal wells, and the ratio obtained was calculated as the cell survival rate. Cell survival (%) = (Absorbance intensity of test sample/ Absorbance intensity of control) ×100%.

## Fluorescence confocal imaging

HepG2 cells were seeded in 15 mm glass-bottomed dishes and cultured for 24 h. The medium was then changed into a serum-free medium containing various concentrations of $H_2O_2$ (0, 0.5, 1, or 2 mM). After a further incubation time of 1, 2, or 4 h, cells were washed with PBS three times. Then, these cells were incubated with the probe (5 μM) in serum-free medium for 30 min. After three times of washing with PBS, fresh medium without serum was added into the wells, and fluorescence images were recorded on a Leica TCS SP8 confocal micropscope using Leica Application Suite X software. The ∞/0.17/OFN25/E, HC PL APO, 63×/1.40 OIL CS2 objective len was used. **F-Tz1** and **F-Tz2** channel: $\lambda_{ex} = 405$ nm, $\lambda_{em} = 410–550$ nm, **F-Tz3** and **F-Tz4** channel: $\lambda_{ex} = 405$ nm, $\lambda_{em} = 450–600$ nm, **F-Tz5** and **F-Tz6** channel: $\lambda_{ex} = 405$ nm, $\lambda_{em} = 410–550$ nm, **F-Tz7** channel: $\lambda_{ex} = 488$ nm, $\lambda_{em} = 495–600$ nm, **F-Tz8** channel: $\lambda_{ex} = 552$ nm, $\lambda_{em} = 560–650$ nm, **F-Tz9** channel: $\lambda_{ex} = 488$ nm, $\lambda_{em} = 500–600$ nm. Each experiment was performed three times. Three frames were taken each time for a single focal plane without Z-stack by random selection. For various conditions in one experiment, the same microscope parameters were used to keep the background signal constant. Images were analyzed via the software "-ImageJ" to quantify the fluorescence intensity by densitometry. Briefly, the cellular cytoplastmic regions were outlined freehand and each cell fluorescence measured manually. Mean gray value was used for measuring the fluorescence intensity. No background subtraction was used. No gaussian blur filter was applied. The measurements of the raw data were pooled across various conditions and across the three parallels in one experiment.

For the pretreatment of Tiron or TEMPO, HepG2 cells were seeded in 15 mm glass-bottomed dishes and cultured for 24 h. Then cells were first treated with Tiron (100 μM) or TEMPO (300 μM) in serum-free DMEM for 1 h. The medium was then changed into fresh medium without serum containing Tiron (100 μM) or TEMPO (300 μM) together with $H_2O_2$ (2 mM). After a further incubation time of 2 h, cells were washed with PBS three times and stained with the probe as above described. (**F-Tz1** channel: $\lambda_{ex} = 405$ nm, $\lambda_{em} = 410–550$ nm, **F-Tz4** channel: $\lambda_{ex} = 405$ nm, $\lambda_{em} = 450–600$ nm, DCFHDA channel: $\lambda_{ex} = 488$ nm, $\lambda_{em} = 500–550$ nm, DHE channel: $\lambda_{ex} = 552$ nm, $\lambda_{em} = 560–630$ nm).

For the cell experiment stimulated with paraquat, HepG2 cells were seeded in 15 mm glass-bottomed dishes and cultured for 24 h. The medium was then changed into a serum-free medium containing various concentrations of paraquat (0, 0.5, 1, or 3 mM). After 24 h, cells were washed with PBS three times. Then, these cells were incubated with the probe **F-Tz4** (5 μM) in serum-free medium for 30 min. After three times of washing with PBS, fresh medium without serum was added into the wells, and fluorescence images were recorded on a Leica TCS SP8 confocal micropscope. **F-Tz4** channel: $\lambda_{ex} = 405$ nm, $\lambda_{em} = 450–600$ nm.

For the cell experiment stimulated with EGF, A549 cells were seeded in 96-well culture plates with edge-well PBS replenishment and incubated in a 37 °C incubator with 5% $CO_2$ for 12 h. For DPI inhibition experiment, the cells were divided into three groups. The control group and EGF group were treated with 0 or 0.5 μg/mL of EGF in serum-free DMEM for 30 min. The DPI group was pretreated with NOX inhibitor DPI (5 μM) in serum-free DMEM for 30 min, and then the medium was changed into fresh medium without serum but containing DPI (5 μM) together with 0.5 μg/mL EGF for 30 min. For the VAS2870 inhibition experiment, cells were divided into four groups. The control group and EGF group were treated with 0 or 0.5 μg/mL of EGF in serum-free DMEM for 30 min. The VAS2870 group was pretreated with NOX inhibitor VAS2870 (10 or 20 μM) in serum-free DMEM for 60 min, and then the medium was changed into fresh medium without serum but containing VAS2870 (10 or 20 μM) together with 0.5 μg/mL EGF for 30 min. The cells were washed with PBS three times and incubated with probe **F-Tz4** (5 μM) in serum-free medium for 30 min. After three times of washing with PBS, fresh medium without serum was added into the wells, and fluorescence images were recorded on a Leica TCS SP8 confocal micropscope. **F-Tz4** channel: $\lambda_{ex} = 405$ nm, $\lambda_{em} = 450–600$ nm.

For the organelle tracker co-localization imaging experiment, HepG2 cells were seeded in 15 mm glass-bottomed dishes and cultured for 24 h. Then cells were treated with $H_2O_2$ (0.5 or 2 mM) in a serum-free medium for 2 h. After that, cells were washed with PBS three times, Then, the cells were stained with **F-Tz4** (5 μM) in serum-free medium for 30 min. After washing with PBS three times, the cells were further stained with the commercial organelle tracker (50 nM Mito-Tracker Red CMXRos, 50 nM Lyso-Tracker Red, or 5 μM DHE for nucleus) in serum-free medium for 15 min. Fluorescence images were recorded on a Leica TCS SP8 microscopy (**F-Tz4** channel: $\lambda_{ex} = 405$ nm, $\lambda_{em} = 450–580$ nm, The Mito-Tracker Red CMXRos channel: $\lambda_{ex} = 552$ nm, $\lambda_{em} = 580–700$ nm, The Lyso-Tracker Red channel: $\lambda_{ex} = 552$ nm, $\lambda_{em} = 580–700$ nm, The DHE channel: $\lambda_{ex} = 552$ nm, $\lambda_{em} = 560–630$ nm).

## Detection of 2-hydroxyethidium in HepG2 cells by LC-MS

HepG2 cells ($6 \times 10^5$) were seeded in 60 mm diameter dishes and cultured for 24 h. The medium was then changed into a serum-free medium containing various concentrations of $H_2O_2$ (0, 0.5, 1, or 2 mM). For the TEMPO group, cells were first treated with TEMPO (300 μM) in serum-free DMEM for 1 h. The medium was then changed into fresh medium without serum containing TEMPO (300 μM) together with $H_2O_2$ (2 mM). After a further incubation time of 2 h, cells were washed with PBS three times. Then, these cells were incubated with DHE (10 μM) in serum-free medium for 30 min. After 30 min, remove the medium and wash the cells with ice-cold DPBS. Then scrape the cells in 1 ml of ice-cold DPBS. Transfer the cell suspension into 1.5 mL Eppendorf tube and centrifuge (5 min × 94 g, 4 °C). Remove the supernatant and then add 150 μL of ice-cold DPBS containing 0.1% (v/v) Triton X-100. Draw the mixture in and out of the insulin syringe (ten times) to lyse the cells and centrifuge (5 min × 94 g, 4 °C). Transfer 100 μL of the lysate supernatant into the tube containing 100 μL of 0.2 M HClO$_4$ in ice-cold MeOH. Votex 10 s and place in ice for 2 h to allow protein precipitation. During this time, transfer 2 μL of the same lysate supernatant to quantify protein used BCA Kit. After 2 h, pellet the protein precipitate by centrifugation (30 min × 376 g, 4 °C). Transfer 100 μL of resulting supernatant to the tube containing 100 μL of 1 M PBS (pH 2.6). Votex 5 s and remove the KClO$_4$ precipitate by centrifugation (15 min × 94 g, 4 °C). Transfer 150 μL of supernatant into LCMS vial equipped with 200-μL conical glass insert and analyze by LCMS (Acquisition Mode: SIM). The calculated and detected $m/z$ values are as follows: DHE: $[M + H]^+$: 316; [2-OH-E]$^+$: 330; [E]$^+$: 314. The specific product of the reaction between DHE and superoxide is 2-hydroxyethidiun (2-OH-E$^+$), and ethidium E$^+$ is produced as a nonspecific product. For the 2-OH-E$^+$ quantification, the peak area detected at $m/z = 330$ was used. The calculation of the amount of 2-OH-E$^+$ = peak area/protein concentration (mg/mL). We also used cell-free samples as controls to guarantee that no 2-OH-E$^+$ was produced during the procedures.

## In vivo imaging with F-Tz4

Male adult C57B6/L mice (8 weeks old) were obtained from SLAC ANIMAL Company, Shanghai. The mice were first acclimatized and fed for one week. The mice were randomly divided into two groups, one for the sham-operated group and the other for the I/R injury group. Before surgery, the mice were opened and intracardially injected with **F-Tz4** probe. To assess the myocardial injury, mice were subjected to 45 min of myocardial ischemia followed by 15 min of reperfusion. The sham-operated group underwent the same procedure. Afterward, the heart was removed, frozen, and placed under a slide, and images were

acquired using fluorescence confocal microscopy. Fluorescence intensity was quantified by ImageJ software.

## High-content screening methods

H9C2 cells were cultured in high glucose Dulbecco's modified Eagle medium (DMEM, Gibico) supplemented with 10% fetal bovine serum (FBS, PAN), and 1% antibiotics (100 U/ml penicillin and 100 μg/ml streptomycin) at 37 °C and 5% $CO_2$. Cells were carefully harvested and split when they reached 80% confluence to maintain exponential growth.

For drug screening, H9C2 cells were first seeded into a 96-well black plate with a clear bottom at a density of 5000/well. After culturing the cells for 24 h, the compounds were administrated at 25 μM, and incubated with the cells for 24 h in DMEM. Then the medium was removed, and cells were treated with 150 μM $t$BHP in DMEM for 2 h. After removing the medium, cells were washed with DMEM three times, then stained with 1 μM Nuclear Red in DMEM for 10 min. The medium was removed and cells were washed with DMEM three times. Then the cells were stained with F-Tz4 (1 μM) in DMEM for 30 min. All Fluorescent images of cell were captured with an ImageXpress Micro Confocal High-Content Imaging System (F-Tz4 channel: $\lambda_{ex} = 405$ nm, $\lambda_{em} = 450$–580 nm, the nuclear red channel: $\lambda_{ex} = 552$ nm, $\lambda_{em} = 580$–700 nm) with the 40× air objective len (S PLAN FLUOR ELWD, 0.60NA). Each condition was run in three replicates. The image analysis software of the high-content imaging system (MetaXpress PowerCore) was used to calculate the green fluorescence intensity in the cell fluorescence images and count the number of cells according to the number of fine nuclei. The inhibition rate was calculated according to the following formula: inhibition rate = $[(I_M - I_{compound})/I_M] \times 100\%$. $I_{compound}$ represents the average fluorescence intensity of green fluorescence per cell in the administered group, $I_M$ represents the average fluorescence intensity of green fluorescence per cell in the administered group.

## Protective effect of coprostanone against mice myocardial I/R injury

Male adult C57BL/6 mice (8 weeks old) were obtained from SHANGHAI SLAC ANIMAL CO. LTD (Certificate No.: SCXK [Hu] 2017-0005). The mice were first adaptive feeding for one week. According to the random number table method, 132 mice were randomly divided into sham operation group, I/R group, 5αCh3 low dose group (50 mg/kg) and 5αCh3 high dose group (100 mg/kg). 5αCh3 was dissolved with 5% Macrogol (15)-Hydroxystearate (AA Blocks)-Saline. The drug was administered by gavage once a day for 3 days. The sham-operated and I/R groups were given equal amounts of the blank solvent.

After 3 days pretreatment with or without 5αCh3, mice were subjected to myocardial ischemia and reperfusion injury by occlusion of the left anterior descending coronary artery for 45 min followed by 24 h reperfusion. The sham-operated group underwent the same procedure without ligation of the left anterior descending coronary artery. All the mice were anesthetized with tribromoethanol (Sigma-Aldrich) at a dose of 150 mg/kg before surgery.

After 24 h, the survived mice for TTC staining were injected with Evans blue dye (Sigma-Aldrich) through the aortic arch retrogradely into the heart to depict the area at risk. Hearts were rapidly excised and transferred to −80 °C. Then the hearts were cut into 5 pieces and incubated in 1.0% 2,3,5-triphenyltetrazolium chloride (Sigma-Aldrich) solution at 37 °C for 10 min. The TTC staining solution was aspirated dry, and the sections were fixed in 4% Paraformaldehyde Fix Solution (Sangon Biotech) and photographed 2 h later. The area of Infarct size (INF), left ventricular(LV), and area at risk (AAR) were measured using ImageJ. The ratio of INF/LV (%), AAR/LV(%), and INF/AAR(%) were calculated.

The remaining saline-washed hearts of mice were divided into two parts, the upper part of the heart was reserved for immunofluorescence staining and the lower part was reserved for western blot analysis.

## Testing the effects of coprostanone on hemodynamics in mice

41 Mice were randomly divided into four groups, including sham, sham + 5αCh3 (100 mg/kg), I/R, and I/R + 5αCh3 (100 mg/kg) groups. After 3 days of pretreatment with 5αCh3 or vehicle, I/R and sham operations were performed. After 45 min ischemia followed by 24 h reperfusion, all the survived mouse were anesthetized and then micro-cannulated with a 1.4F microcatheter (Millar Instrument Inc, USA) and measured using a Powerlab multichannel physiological recorder. Briefly, the right common carotid artery (CCA) was carefully isolated and adequately exposed, followed by transiently blocking the proximal end of CCA using a microvascular clip. After ligation and traction of the distal end of CCA, a small hole was cut in the right CCA to allow the microcatheter to insert. Loosening the clip and gently advancing the microcatheter to reach the left ventricle. The key parameters, including left ventricular systolic pressure (LVSP), left ventricular end-diastolic pressure (LVEDP) and the maximum and minimum rates of left ventricular pressure changes (dp/dt max and dp/dt min), were automatically recorded. 1-3 mice in each group failed to record the hemodynamic parameters because of the failure of micro-cannulation.

## Western blot

The heart tissues of mice collected from sham-operated group, I/R group, 5αCh3 low dose group (50 mg/kg), 5αCh3 high dose group (100 mg/kg) with four mice in each group were removed from −80 °C and thawed on ice. Weigh and cut the tissues into homogenization tubes, add a small amount of small magnetic beads, and use a homogenizer to obtain a tissue homogenate. After centrifugation, the supernatant was taken and BCA assay was used to quantify the protein. Protein samples were mixed with loading buffer and reducing agent, then separated on 10% Bris-tris gels and blotted on PVDF membranes. Subsequently, membranes were closed in closure buffer (5% Non-Fat milk) and incubated overnight with primary antibodies to NRF2 (Cat. no. AF7623, Beyotime), SOD2 (Cat. no. 24127-1-AP, Proteintech), HO-1(Cat. no. 10701-1-AP, Proteintech), and β-tubulin (Cat. no. 10094-1-AP, Proteintech) (1:1000 in primary antibody diluent), washed 3 times, and then incubated with HRP-conjugated anti-Rabbit secondary antibody (Cat. no. A0208, Beyotime) (1:2000 in 5%BSA-TBST) at room temperature for 1 h. ECL chemiluminescent reagents were used to show the blots. These bands were exposed by Bio-Rad ChemiDoc XRS.

## Immunofluorescence staining

After 24 h reperfusion, the mice were anesthetized and the hearts ($n = 4$ for each group) were extracted and fixed in 4% paraformaldehyde at 4 °C for at least 48 h. Briefly, the cardiac tissues were dehydrated, embedded, and cut into 4 μm-thick slices. Then, the paraffin sections were dewaxed and rehydrated followed by treated with EDTA Antigen Repair Solution (Servicebio, China). After blocking with bovine serum albumin solution (BSA, Sangon Biotech, China) for 30 min at room temperature, the slices were incubated overnight at 4 °C with primary antibodies against NRF2 (1:5000, Cat. no. AF7623, Beyotime)/SOD2(1:5000, Cat. no. 24127-1-AP, Proteintech)/HO-1 (1:6000, Cat. no. 10701-1-AP, Proteintech). Subsequently, the corresponding HRP conjugated Goat Anti-Rabbit IgG (H + L) (1:500, Cat. no. GB23303, Servicebio) was applied to treat the slices at room temperature for 50 min. After incubating with FITC reagent at room temperature in dark for 10 min, the sections were treated with Tris-EDTA followed by blocking. Next step, the slices continued to be incubated overnight at 4 °C with anti-cTnT antibody (1:300, Cat. no. GB11364, Servicebio) followed by treated with Cy3 conjugated Goat Anti-Rabbit IgG (H + L) (1:300, Cat. no. GB21303, Servicebio) for 50 min at room temperature. Cell nuclei was stained with 4′,6-diamidino-2-phenylindole (DAPI) for 10 min at room temperature in dark. The images

were obtained and observed using a Orthofluorescence microscope (NIKON ECLIPSE C1, Japan). For the cells with expression of NRF2, HO-1 and SOD2, the cells were presented with green color in cytoplasm, blue color in nucleus and the red color represented cTnT.

## HPLC analysis procedures

To record the reactivity of **Tz1-Tz4** towards $O_2^{\cdot-}$, each compound stock solution (50 mM, in DMSO) was diluted in MeCN to make a 100 μM solution. Then $O_2^{\cdot-}$ (0–20 eq) was added. The mixture was stored at ambient temperature for 30 min, followed by HPLC analysis. The peak area of the remaining residue was used to calculate the conversion rate. The detailed liquid chromatography methods were reported in the Supplementary Information. To record the reactivity of **Tz1** towards various analytes, each analyte was added to an aliquot of **Tz1** in DMSO. After 30 min, the mixture was diluted with a mixed solution of MeCN and PBS (1:1, v/v, PBS of pH 7.4, 10 mM). Then the mixture was analyzed by HPLC. The peak area corresponding to 2,5-diphenyl-1,3,4-oxadiazole was recorded, which was normalized to the $KO_2$ group to calculate the normalized yield of the oxadiazole product. The peak area corresponding to **Tz1** residue was also recorded, which was normalized to the blank group to calculate the residue of **Tz1**.

## Optical response analysis

To measure the absorption spectra, the stock solution of **F-Tz1-F-Tz9** was diluted with MeCN containing 1% 18-crown-6. $O_2^{\cdot-}$ was added to this solution. After 30 min of incubation, the mixture was diluted with PBS buffer (10 mM, pH 7.4) and then measured. The final concentration of the probe was 20 μM while that of $O_2^{\cdot-}$ was 400 μM. To record the fluorescence spectra, various doses of $O_2^{\cdot-}$ (0–20 eq) were added to probe **F-Tz1-F-Tz9** solutions in MeCN containing 1% 18-crown-6. After 30 min of incubation, the mixture was diluted with PBS buffer (10 mM, pH 7.4) to make sure that the final probe concentration was 5 μM. The mixture was then measured for fluorescence (**F-Tz1**: $\lambda_{ex}/\lambda_{em} = 323/470$ nm; **F-Tz2**: $\lambda_{ex}/\lambda_{em} = 384/460$ nm; **F-Tz3**: $\lambda_{ex}/\lambda_{em} = 350/530$ nm; **F-Tz4**: $\lambda_{ex}/\lambda_{em} = 385/510$ nm; **F-Tz5**: $\lambda_{ex}/\lambda_{em} = 405/445$ nm; **F-Tz6**: $\lambda_{ex}/\lambda_{em} = 380/505$ nm; **F-Tz7**: $\lambda_{ex}/\lambda_{em} = 445/540$ nm; **F-Tz8**: $\lambda_{ex}/\lambda_{em} = 565/585$ nm; **F-Tz9**: $\lambda_{ex}/\lambda_{em} = 488/600$ nm). To determine the quantum yields, quinine sulfate ($\Phi_{standard} = 0.577$ in 0.1 M $H_2SO_4$) and fluorescein ($\Phi_{standard} = 0.95$ in 0.1 M NaOH) were used as standards according to a published method. For the probe, fluorescein, and quinine sulfate, the absorbance spectra were measured within an absorbance range of 0.01 to 0.05. The quantum yield was calculated according to the equation:

$$\Phi_{sample} = \Phi_{standard} \frac{\sum F_{sample}}{\sum F_{standard}} \frac{Abs_{standard}}{Abs_{sample}} \left(\frac{n_{sample}}{n_{standard}}\right)^2 \qquad (1)$$

where $\Phi$ is the quantum yield, $\Sigma F$ is the integrated fluorescence intensity, Abs is the absorbance at the excitation wavelength, and n represents the refractive index of the solvent.

To test the selectivity of the probes, **F-Tz1-F-Tz9** was diluted with PBS buffer (10 mM, pH 7.4) to make a solution of 5 μM. Aliquots of this solution were then treated with various analytes whose stock solutions was prepared according to the methods described in the Supplementary Information. The volume change (ca. 1‰) caused by the addition of analytes could be negligible. After incubating at room temperature for 30 min, the fluorescence spectrum was collected. All the fluorescence and absorption spectra data were processed via the software Origin 2021.

## Reduction potentials and cyclic voltammograms

**Tz1-Tz3** (1 mmol) was dissolved in 10 ml of 0.1 M $Et_4NClO_4$ solution of $CH_3CN$. Reduction potentials and cyclic voltammograms were recorded on an Electrochemical workstation (CHI660E, produced by Shanghai YueCi Electronic Technology Company) using glassy carbon

as working, platinum as counter, and Ag/AgCl as reference electrode, respectively.

## Computational methods

Quantum chemical calculations based on density functional theory (DFT) and time-dependent density functional theory (TD-DFT) were employed to rationalize the fluorogenicity of tetrazine-tethered probes. All structure optimizations were performed without constraints using the ωB97XD functionals and def2SVP basis set in the ground states. Solvation effects (in water) were taken into account using the SMD model. The vertical excitation energies of all molecules were calculated using linear solvation formalism at TDDFT-ωB97XD/def2SVP. All DFT and TD-DFT calculations were carried out using Gaussian 16A.

## Statistics and reproducibility

All the statistical analysis was perfomed with Graphpad Prism 8.0 software. The two-tailed unpaired Student's t-test and one-way Anova followed by Dunnett's multiple comparisons test or uncorrected Fisher's LSD multiple comparisons test were used for data statistical handled. $P$ value <0.05 was considered as statistically significant. All the cell and animal experiments was performed at least three biologically independent times.

## Reporting summary

Further information on research design is available in the Nature Portfolio Reporting Summary linked to this article.

## Data availability

All data generated in this study are provided in the Supplementary Information and Source data file. Source data are provided with this paper.

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

## Acknowledgements

This study was supported by grants from the National Natural Science Foundations of China (82173941 to Y.W., 21778048 and 22077112 to X.L.). Y.W. was supported by Innovation Team and Talents Cultivation Program of National Administration of Traditional Chinese Medicine (No: ZYYCXTD-D-202002). X.L. was supported by the National Program for Support of Top-notch Young Professionals (grant 2021). The authors also appreciate the supports from the Fundamental Research Funds for the Central Universities (226-2022-00226) and ZJU PII-Molecular Devices JOINT LABORATORY, SUTD-ZJU IDEA grant [SUTD-ZJU (VP) 201905, to Xiaogang L.], and the National Supercomputing Centre (Singapore).

## Author contributions

X. Li and Y.W. conceived and designed the project. X.J. synthesized the compounds, characterized the photophysical properties of the probes, and conducted the cell imaging experiments. M.L. performed the high-content screening and analyzed the data. Y.L.W. and Y.C.W. performed the animal experiments. C.W. and T.S. performed the computational calculation under the guidance of X. Liu. L.S. synthesized part of the intermediates. X.J. and M.L. drafted the manuscript. X. Li, Y.W., and X. Liu revised it. All authors read and approved the final manuscript.

## Competing interests

The authors declare no competing interests.
