## [Peer Review File · Nature Communications]

REVIEWER COMMENTS

Reviewer #1 (Remarks to the Author):

The present paper reports the development of a novel fluorescent probe for superoxide, its use in the cellular monitoring of superoxide, its use in the screening for natural compounds which reduce the formation of superoxide, and finally use of one of the compounds (coprostanone) which were identified by the screening in attenuating myocardial ischemia/ reperfusion injury in an anesthetized mouse model.

My competence relates to myocardial ischemia/reperfusion injury and I therefore restrict my comments to that:

The value of H9C2 cells in cardioprotection research is very limited, in that these cells are largely dedifferentiated and noncontractile (Basic Res Cardiol 113,2018,39 and 116,2021,52). Their use for larger scale screening is acceptable, but the authors must nevertheless acknowledge that cardiac contraction per se generates ROS (Cardiovasc Res 71,2006,374-82) which they are missing in H9C2 cells.

The cardioprotection studies in the mouse model used appropriate methods to measure infarct size which is indeed the most robust endpoint of cardioprotection. However, a lot of pertinent information is missing: anesthesia, the exact induction of ischemia by occlusion of which coronary artery, hemodynamics to judge side effects of coprostanone which impact on infarct size. The n-values for the TTC data are too low for robust data, in mouse models n-values of 8-10 are recommended and data should be presented as scatter plots.

The Western data of NRF2, HO-1 and SOD2 show a single, cut-out Western example without statistics. These data are anecdotal and associative at best, and there is no proof for causality between these Western data and the observed infarct size reduction by coprostanone.

Finally, a number of small molecules which attenuate ROS formation and myocardial ischemia/reperfusion injury in small rodent models exist (vitamin C, MPG, NAC) but there has been no successful translation of their use to clinical benefit for patients with acute myocardial infarction so far, and the same is true for SOD and catalase- this limitation must be acknowledged.

Reviewer #2 (Remarks to the Author):

Jiang and co-workers report on the development of superoxide selective activity-based probes based on 1,2,4,5-tetrazines. They present a rigorous study on structure-activity relationships within an extensive panel of probe derivatives and apply these in a high-throughput fluorescence microscopy assay of 233

natural product derivatives to discover new superoxide modulators. Overall, the chemical aspects of this paper are detailed and thoughtfully presented, however I do have concerns about the microscopy experiments which are elaborated below. The need for 2 mM hydrogen peroxide in live cells to visualize a fluorescence response hampers the usefulness of this system but given the extreme importance of identifying highly selective superoxide probes, this manuscript should provide a solid foundation for the discovery of more sensitive fluorophores. Thus, my recommendation is that this paper should be accepted if the authors are able to address the following concerns:

1.) The authors are encouraged to cite some new key references in the ROS literature:

Murphy, M. P. et al. "Guidelines for measuring reactive oxygen species and oxidative damage in cells and in vivo" *Nat. Metab.* 2022, 4, 651-662.

Sies, H. et al. "Defining roles of specific reactive oxygen species (ROS) in cell biology and physiology" *Nat. Rev. Mol. Cell Biol.* 2022, 23, 499-515.

2.) Rf values for compounds purified via silica gel chromatography should be reported.

3.) For all the cell imaging experiments it is unclear what objective lens (air, water, or oil) and numerical aperture was used.

4.) Very little information about how fluorescence quantification was carried out is given. Unfortunately, "Fluorescence intensity was quantified via the software "-ImageJ"." Is not sufficient detail. Was there background subtraction used? Z-stacks? How was thresholding carried out? Was a gaussian blur filter applied? These are all the bare minimum. Ideally, if supplied with the raw data, another independent laboratory should be able to obtain the same results.

5.) There are aspects of Figure 1 that can be clarified to better help the reader. It would help to add the actual VEA numbers on top of the bars in 1C and include the reduction potential of the superoxide radical anion somewhere in the figure.

6.) Figure 1F is confusing, it may be better to only show the O1 HPLC yield, showing the Tz1 residue in the same graph only serves to obfuscate the point.

7.) Figure 1H, the conversion rate of Tz1 and Tz4 is also confusing as I am unsure what the green dots denote. It may be clearer to show a simple bar graph with error bars if the green dots denote replicates.

8.) Figure 2E represents what I think is an extremely important experiment. However, it is difficult to see the data due to a mixture of resolution and the amount of data relative to the size of the overall figure. The authors should consider increasing the size of Figure 2C,D, and E, and making an entire page figure by shifting F and G down further or splitting it into a separate figure.

9.) Brightfield images for all confocal data presented should be shown.

10.) To visualize a fluorescent signal, the authors needed to treat live cells with an excessive (2 mM) amount of H₂O₂. As such, the authors should perform in vitro experiments using 2 mM H₂O₂ in the presence of their Tz probes to demonstrate a lack of reactivity at that concentration. While the LOD of the Tz probes in vitro is impressive (nm), this doesn't seem to be translated in cellulo potentially

indicating that the kinetics are not favorable to capture superoxide in a relevant time scale at low concentrations.

11.) Along these lines, are the authors able to capture endogenous superoxide generation (ie with cell treatment using paraquat or EGF).

12.) Cell viability with F-Tz compounds was performed, but it would be beneficial to perform cell viability experiments with the generated oxadiazole derivatives.

13.) Figures S46 and S47 are concerning as there appears to be significant cell death and detachment at the 'optimized' levels of tBHP and F-Tz4 treatment. Thus, it would be beneficial to provide all the images used for the fluorescence intensity measurement of the key experiment shown in 5D somewhere in the SI. Additionally, it seems that pdf compression has significantly reduced the resolution of the images, whether this was on the journal's end or the author's, it should be remedied.

Reviewer #3 (Remarks to the Author):

This research is related to diagnosis of superoxide ion generation in mammal cells. 1,2,4,5-Tetrazine-tethered as fluorescent probes. despite the effort invested to come up with this work, the structure of the article is limited and requires the following modifications:

1. The discussion section is rather limited and looks like a review rather than a critical discussion on the results obtained. Merging both results and discussion sections would be preferable.

2. The tenses used did not follow the standard writing commonly used in drafting research articles. As a result, the article needs a proofread by a native

English speaker.

3. To verify the superoxide ion detection, more details on the electrochemical approach should be added (e.g. cyclic voltammetry).

4. The question that must be addressed carefully is how this probe can distinguish the superoxide radical from other ROS species.

5. It is highly recommended to report the shortcomings of using this method and how the authors could overcome. In addition, it should report the unavoidable defects in this method.

6. EPR spin-trapping technique should be used for comparison purpose.

7. A table of comparison with other fluorescent probes and other superoxide detection methods should be initiated and added to discussion section.

RESPONSES TO REVIEWER COMMENTS

Reviewer #1:

1. *The value of H9C2 cells in cardioprotection research is very limited, in that these cells are largely dedifferentiated and noncontractile (Basic Res Cardiol 113,2018,39 and 116,2021,52). Their use for larger scale screening is acceptable, but the authors must nevertheless acknowledge that cardiac contraction per se generates ROS (Cardiovasc Res 71,2006,374-82) which they are missing in H9C2 cells.*

Thanks for this comment, and we appreciate the reviewer for this support.

We acknowledge that contracting cardiomyocytes produce ROS. However, this physiological level of ROS should be significantly below the pathological levels under oxidative stress. To make a confirmation, we have stained primary mice cardiomyocytes without or with tBHP (30 μ M) pretreatment with probe F-Tz4. Oxidative stress significantly enhanced cellular probe fluorescence intensity, while this increase could be compromised by the pretreatment of cells with coprostanone (Figure S58 in ESI Page S64), suggesting that the imaging and screening results should be reproducible in primary mice cardiomyocytes. For your quick reference, we have also attached the figure below.

Specifically, we used the H9C2 cells for the high-content screening experiments due to their ready availability. Since in the end, the activity of the potential lead compound should be confirmed in animal models, we think that the use of H9C2 cells should be acceptable.

Figure S58. Protective effect of 5 α Ch3 in tBHP-injured primary cardiomyocytes. A) Fluorescence images of primary cardiomyocytes cells stained with 10 μ M of F-Tz4 for 30 min respectively. Before staining, cells were intact (control) or pretreated with coprostanone (10 or 25 μ M) for 24 h. All groups were then treated with tBHP (30 μ M) except the cells in control group for 2 h, and then stained with F-Tz4 for 30 min. Images were acquired using an ImageXpress Micro Confocal High-Content Imaging System (Molecular Devices), with a 40 \times PlanFluor objective. Scale bar: 20 μ m. B) The statistically quantified data on the cellular fluorescence intensity in A. The data

were the mean \pm SD and were normalized to the control group. $n = 3$ (3 wells or slides/group). All data were analyzed using one-way ANOVA and data were expressed as means \pm SD, ** $p < 0.01$, *** $p < 0.001$, versus tBHP group.

- 2. The cardioprotection studies in the mouse model used appropriate methods to measure infarct size which is indeed the most robust endpoint of cardioprotection. However, a lot of pertinent information is missing: anesthesia, the exact induction of ischemia by occlusion of which coronary artery, hemodynamics to judge side effects of coprostanone which impact on infarct size. The n-values for the TTC data are too low for robust data, in mouse models n-values of 8-10 are recommended and data should be presented as scatter plots.*

We appreciate this comment.

All the mice were anesthetized with tribromoethanol (Sigma-Aldrich) at a dose of 1.5 mg/kg before surgery. Myocardial ischemia/reperfusion injury was induced by occlusion of the left anterior descending coronary artery for 45 min followed by 24 h reperfusion. These related details for this experiment were added in the ESI (Page S23).

To test the effect of coprostanone on hemodynamics, we used a zebrafish model and observed the effect of different concentrations of coprostanone on blood flow in zebrafish embryos. As shown in the following figure, concentrations as high as 50 μ M did not affect the blood flow in zebrafish embryos. We assume that coprostanone should have no side effects on the hemodynamics of mice which impact infarct size.

Figure. Effect of coprostanone on blood flow in zebrafish embryos. The zebrafish experiments were conducted according to the guidelines of Animal Ethics Committee of the Laboratory Animal Center, Zhejiang University. Wildtype TU strain, Tg (LCR: EGFP) (22537494) was obtained from the Laboratory Animal Center of Zhejiang University. Zebrafish were maintained following standard protocols (Westerfield, M. The zebrafish book: a guide for the laboratory use of zebrafish (Danio rerio). M. Westerfield, 2007). E3 medium (0.29 g/L NaCl, 0.013 g/L KCl, 0.048 g/L CaCl₂•2H₂O, 0.082 g/L MgCl₂•6H₂O, pH 7.2) was used as the embryo medium. Embryos were obtained through natural spawning. 5αCh3 were diluted to 10, 25 and 50 μ M with E3 medium and used to culture zebrafish embryos for 48 h before placing them under a fluorescent microscope to take videos of the embryonic blood flow. Ten zebrafish embryos were filmed per group.

For the n-values for the TTC data, we have performed additional experiments with more mice. The final biological replicates of the TTC staining experiment were 15-16. The results were updated in Figure 6D and 6E (revised into scatter plots), and raw data for each replicate were also

submitted. According to the data, a steady dose-dependent reduction in infarct size was observed, supporting the protective effect of coprostanone. To further illustrate this protective effect of coprostanone on mouse heart during ischemia-reperfusion injury, we performed immunofluorescence staining of NRF2, HO-1 and SOD2 proteins on mouse heart tissues. The biological replicates of each group were four. The results showed that this compound significantly increased the expression of NRF2, HO-1 and SOD2 in heart tissues in response to oxidative damage during ischemia-reperfusion (Figure S59 in ESI Page S65).

3. *The Western data of NRF2, HO-1 and SOD2 show a single, cut-out Western example without statistics. These data are anecdotal and associative at best, and there is no proof for causality between these Western data and the observed infarct size reduction by coprostanone.*

We appreciate this comment.

Additional experiments were performed to ensure each group contain four biological replicates of Western blot, and the original blots were also submitted as source data. By performing the statistic analysis with two-tailed test (data updated in Figure 6F), the trend of increasing expression of NRF2, HO-1 and SOD2 was obvious in mice pretreated with coprostanone and then subjected to I/R injury, with some demonstrating statistic significance.

Noteworthy, we also carried out immunofluorescence assay on the heart tissues from each group, and further confirmed the upregulation of these antioxidant enzymes in the groups pretreated with coprostanone (ESI, Page S65, Figure S59). For your quick reference, the result on the immunofluorescence assay was attached below.

Figure S59. Representative images of immunofluorescence staining of NRF2, HO-1 and SOD2 in the heart tissues of mice in sham operation group, I/R group, 5αCh3 low dose group (50 mg/kg), 5αCh3 high dose group (100 mg/kg) respectively (n = 4 mice for each group). Green represents the target protein, blue represents the nucleus, and red represents cTnT in these images. Scale bar = 20 μm.

4. *Finally, a number of small molecules which attenuate ROS formation and myocardial*

ischemia/reperfusion injury in small rodent models exist (vitamin C, MPG, NAC) but there has been no successful translation of their use to clinical benefit for patients with acute myocardial infarction so far, and the same is true for SOD and catalase-. This limitation must be acknowledged.

We appreciate the reviewer for raising this point.

It is true that some molecules directly scavenging ROS such as vitamin C, vitamin E, NAC, etc, have failed to give the expected protective results in large clinical trials. The following possible reasons have been proposed. 1) These antioxidants are not robust enough to scavenge the huge overproduction of ROS. 2) Injury by oxidative stress occurs at the very early onset of diseases while these antioxidants are used too late in clinic practice (Nat Med, 2014, 711). 3) Redox signaling has diverse cellular functions while unselectively scavenging ROS by these broad antioxidants may disturb redox signaling (Nat Med, 2014, 711; Nat Rev Mol Cell Biol, 2020, 363). In this way, most of the current antioxidant drug research has switched to the design of antioxidant enzyme mimics, ROS-producing enzyme inhibitors, and agents inducing endogenous antioxidant enzymes (Nat Rev Drug Discov, 2021, 689; J Med Chem 2021, 64, 5252).

In our opinion, the induction of native antioxidant enzymes is especially appealing because this strategy has the potential to re-balance the cellular redox homeostasis. In this context, a cell-based high-content screening model is ideal for this purpose. Specifically, we have confirmed that the lead compound coprostanone in this work protected mice from myocardial ischemia-reperfusion injury by inducing cellular antioxidant enzymes to decrease cellular superoxide levels.

We have added a brief discussion on this point in the main text (Page 15, 21, text in blue).

Reviewer #2:

- 1) The authors are encouraged to cite some new key references in the ROS literature: Murphy, M. P. et al. "Guidelines for measuring reactive oxygen species and oxidative damage in cells and in vivo" Nat. Metab. 2022, 4, 651-662. Sies, H. et al. "Defining roles of specific reactive oxygen species (ROS) in cell biology and physiology" Nat. Rev. Mol. Cell Biol. 2022, 23, 499-515.***

We appreciate the reviewer for the support, and for this suggestion on the references.

These new references are highly related to this work and have been added (Ref 2, 6).

- 2) Rf values for compounds purified via silica gel chromatography should be reported.***

Rf values of the compounds were measured and reported in the ESI.

- 3) For all the cell imaging experiments it is unclear what objective lens (air, water, or oil) and numerical aperture was used.***

The related information was added under the subtitle of "Fluorescence Confocal Imaging" in ESI

(Page S21), and “High-Content Screening Methods” (Page S22).

- 4) *Very little information about how fluorescence quantification was carried out is given. Unfortunately, “Fluorescence intensity was quantified via the software “-ImageJ”.” Is not sufficient detail. Was there background subtraction used? Z-stacks? How was thresholding carried out? Was a gaussian blur filter applied? These are all the bare minimum. Ideally, if supplied with the raw data, another independent laboratory should be able to obtain the same results.***

We are sorry for the negligence on the related information.

For the confocal imaging experiments, each condition was performed three times, and three frames were taken each time without Z-stack by random selection of the imaging region. Cellular fluorescence intensity was analyzed by densitometry using ImageJ, with the pixel intensity of the cellular cytoplasmic regions measured with at least 18 cells per condition. No background subtraction was used. No Gaussian blur filter was applied. The relative fluorescence intensity was normalized to the control group where cells were treated with only the probe but not H₂O₂. The normalized fluorescence intensity was plotted by software Graphpad Prism8.0.2. This information was added under the subtitle of “Fluorescence Confocal Imaging” in ESI (Page S21). All the raw data for plotting were submitted.

For the high-content screening experiment, the cellular probe fluorescence intensity was analyzed by the image analysis software of the high-content imaging system (MetaXpress PowerCore). Cell numbers were counted by this system according to the number of fine nuclei. Related information has been added under the subtitle of “High-Content Screening Methods” in ESI (Page S22).

Statistic methods were also described under the subtitle of “Statistics” in ESI (Page S24), and also briefly described in figure captions.

- 5) *There are aspects of Figure 1 that can be clarified to better help the reader. It would help to add the actual VEA numbers on top of the bars in 1C and include the reduction potential of the superoxide radical anion somewhere in the figure.***

Thanks for this suggestion.

We have added the actual EA values of all oxidization species in Figure 1C. We have also calculated the ionization potential of the superoxide radical anion and inserted this value in the revised Figure 1C as well.

- 6) *Figure 1F is confusing, it may be better to only show the O1 HPLC yield, showing the Tz1 residue in the same graph only serves to obfuscate the point.***

Thanks for this suggestion.

We have moved Tz1 residue to ESI (Figure S7 in Page S36) to make Figure 1F more readable.

- 7) *Figure 1H, the conversion rate of Tz1 and Tz4 is also confusing as I am unsure what the green dots denote. It may be clearer to show a simple bar graph with error bars if the green dots denote replicates.*

We are sorry for this confusion.

The dots represented the values of the conversion rates of the compound upon superoxide treatment, with each experiment performed in three replicates. We have to show all data points to comply with the reporting requirement. To make the figure more readable, we have marked the data attributed to different compounds with different colors and moved the compound caption to the left side.

- 8) *Figure 2E represents what I think is an extremely important experiment. However, it is difficult to see the data due to a mixture of resolution and the amount of data relative to the size of the overall figure. The authors should consider increasing the size of Figure 2C,D, and E, and making an entire page figure by shifting F and G down further or splitting it into a separate figure.*

Thanks for this suggestion.

Figures 2C, D, E have been re-sized as per the suggestion so that data in Figure 2E are now clearer.

- 9) *Brightfield images for all confocal data presented should be shown.*

Thanks for raising this point.

We are sorry for the negligence of omitting the collection of bright field images, although we indeed observed the bright field before collecting the fluorescence images.

To complement this, we have re-performed the experiments corresponding to the data in Figure 3A-B and Figure 4, with both bright field (Figure S45, S46 in Page S56 and Figure S53 in Page S60) and fluorescence images collected. And good reproducibility was obtained. Besides, for the newly-performed experiments such as paraquat or EGF induction, both brightfield and fluorescence images were collected (Figure S40, S41 in Page S54). We believe that these brightfield images should be sufficiently representative to describe the cell morphology in the imaging experiments.

We appreciate this suggestion and will ensure to collect the bright field images in our future work.

- 10) *To visualize a fluorescent signal, the authors needed to treat live cells with an excessive (2 mM) amount of H₂O₂. As such, the authors should perform in vitro experiments using 2 mM H₂O₂ in the presence of their Tz probes to demonstrate a lack of reactivity at that concentration. While the LOD of the Tz probes in vitro is impressive (nM), this*

doesn't seem to be translated in cellulose potentially indicating that the kinetics are not favorable to capture superoxide in a relevant time scale at low concentrations.

Thanks for this comment.

To confirm the inertness of probes F-Tz1-F-Tz4 towards 2 mM H₂O₂, we performed two additional experiments. First, the fluorescence spectra of the probes (5 μM) after the treatment of H₂O₂ (2 mM) at ambient temperature for 30 min were collected. And no significant fluorogenic response was observed (Figure S36 in Page S51). In addition, LC-MS analysis was performed to confirm that the probes (100 μM) were stable in the presence of 2 mM H₂O₂ for at least 30 min (Figure S37 in Page S52). These results should be sufficient to support that H₂O₂ even at a high level of 2 mM still lacks reactivity towards the probes. For your quick reference, the Figures were added below.

In our newly-performed experiments, probe F-Tz4 was sensitive enough to image EGF-induced superoxide burst in live A549 cells (Figure S41, ESI Page S54). This result suggests that the Tz-based probes should favor desirable kinetics to capture cellular low concentrations of superoxide in a relevant short time scale.

Figure S36. Fluorescence spectra of F-Tz1-F-Tz4 after the treatment of 2 mM H₂O₂ or 100 μM KO₂. All probes were used at 5 μM and the reactions were carried out at ambient temperature for 30 min. Then the spectra were recorded for A) F-Tz1, B) F-Tz2, C) F-Tz3, D) F-Tz4.

Figure S37. LC traces of F-Tz1-F-Tz4 before and after the treatment of 2 mM H₂O₂. All probes were used at 100 μM and the reactions were carried out at ambient temperature for 30 min. Then the mixtures were analyzed by LCMS. A) F-Tz1, B) F-Tz2, C) F-Tz3, D) F-Tz4.

11) Along these lines, are the authors able to capture endogenous superoxide generation (ie with cell treatment using paraquat or EGF).

Thanks for raising this point.

Additional experiments were performed as per the suggestion, and probe F-Tz4 was confirmed to capture endogenous superoxide induced by either paraquat or EGF (Figure S40, S41 in Page S54). We also described these experiments and results in the main text (Page 9, text in blue). For your quick reference, the Figures were added below.

Figure S40. F-Tz4 imaging of endogenous superoxide in live HepG2 cells under oxidative stress conditions stimulated by paraquat. A) Confocal microscopy images of HepG2 cells treated with different concentrations (0 - 3 mM) of paraquat for 24 h, and then stained with F-Tz4 (5 μM) for 30 min before imaging (Scale bar: 50 μm). B) Quantified relative mean fluorescence intensity of the cells. Data were mean ± SD (n = 41 – 63 cells) normalized to the negative control group. ***P < 0.001; versus untreated cells.

Figure S41. F-Tz4 imaging of endogenous superoxide in live A549 cells stimulated by EGF. A) Confocal microscopy images of A549 cells treated with 0.5 $\mu\text{g}/\text{mL}$ EGF for 30 min, then stained with F-Tz4 (5 μM) for 30 min before imaging (Scale bar: 50 μm). B) Quantified relative mean fluorescence intensity of the cells. Data were mean \pm SD (n = 18 – 36 cells), normalized to the negative control group. **P < 0.01, ***P < 0.001; versus untreated cells. DPI is a NOX inhibitor and was pre-incubated with cells in serum-free DMEM for 30 min.

12) Cell viability with F-Tz compounds was performed, but it would be beneficial to perform cell viability experiments with the generated oxadiazole derivatives.

Thanks for raising this point.

We have prepared the oxadiazole derivatives (P1-P4) by treating the probes with superoxide. The methods and characterization data were described in the ESI (Scheme S6 in Page S12). These oxadiazoles were then tested for their cytotoxicity in both H9C2 and HepG2 cells. The results showed that no significant cytotoxicity was observed when the cells were incubated with the compounds for 24 h. Results were shown in Figure S35 in ESI Page S50, and also added below for your quick reference.

Figure S35. Cytotoxicity of F-Tz1-F-Tz4 and their oxadiazole-derivatives by MTT assay. A, C) Cytotoxicity assay of probes (A) and oxadiazoles (C) against H9C2 cells. After co-incubating different concentrations of compounds with H9C2 cells for 24 h, the cell viability was evaluated by MTT assay. B, D) Cytotoxicity assay of probes (B) and oxadiazoles (D) against HepG2 cells. After co-incubating different concentrations of probes with HepG2 cells for 24 h, the cell viability was evaluated by MTT assay. n = 3 (3 wells or slides/group, mean \pm SD).

13) Figures S46 and S47 are concerning as there appears to be significant cell death and detachment at the 'optimized' levels of tBHP and F-Tz4 treatment. Thus, it would be beneficial to provide all the images used for the fluorescence intensity measurement of the key experiment shown in 5D somewhere in the SI. Additionally, it seems that pdf compression has significantly reduced the resolution of the images, whether this was on the journal's end or the author's, it should be remedied.

Thanks for raising this point.

We have re-performed the experiments corresponding to the original Figure S46, S47 which are now updated as Figure S54, S55 in ESI Page S60, S60. The results indicated that the optimized levels of tBHP and F-Tz4 treatment caused little cell death.

All the images used for the fluorescence intensity measurement of the experiment shown in original Figure 5D (now updated as Figure 6B) were provided in Figure S57 (ESI Page S63).

Besides, all the data were organized in PowerPoint. We have adjusted the output resolution of the images to be 300 dpi to improve the readability.

Reviewer #3:

1. The discussion section is rather limited and looks like a review rather than a critical discussion on the results obtained. Merging both results and discussion sections would be preferable.

Thanks for raising this point.

We have revised the discussion section, by merging the results and discussion as per the suggestion. Specifically, the design of the probes, data supporting their specificity towards superoxide, and limitations were discussed by merging the results. Revisions were marked as text in blue.

2. The tenses used did not follow the standard writing commonly used in drafting research articles. As a result, the article needs a proofread by a native English speaker.

Thanks for this suggestion.

We have had the manuscript read and revised for the language by a native speaker in USA.

3. *To verify the superoxide ion detection, more details on the electrochemical approach should be added (e.g. cyclic voltammetry).*

Thanks for this comment.

Two electrochemical approaches were employed to verify superoxide detection.

First, we calculated the electron affinity (EA) of various N-substituted benzene rings and the ionization potential (IP) of the superoxide radical anion, finding that this IP is smaller than the EA of Tz (Figure 1C). Detailed methods for calculation were described in the ESI (Page S20) under the subtitle of “Computational methods”.

Second, we measured the reduction potentials of Tz1-Tz3 by cyclic voltammetry, and found that these values positively correlated with the conversion rates of the compounds towards superoxide treatment (Figure 1H). Details on this experiment was added in the ESI under the subtitle of “Reduction Potentials and Cyclic Voltammograms” in Page S20, and data were shown in Figure S12, Page S39 in ESI.

4. *The question that must be addressed carefully is how this probe can distinguish the superoxide radical from other ROS species.*

Thanks for this comment.

The Tz-based probes were designed to distinguish superoxide radical from other ROS by utilizing the reducibility of the superoxide radical; while other ROS tend to grab electrons from surroundings rather than to donate. We have calculated the ionization potential (IP) of the superoxide radical to be 3.156 eV, well smaller than the electron affinity (3.326 eV) of Tz, which makes the Tz-based probes selective towards superoxide. Related values have been added in the main text.

Specifically, we have the following experimental results to support the selectivity of the Tz-based probes for superoxide. 1) Solution-based experiments showed that among the tested species, only superoxide could trigger on the conversion of the tetrazines to oxadiazoles (Figure 1F), and trigger on the fluorogenic response of the probes (Figure 2E, 4B). 2) Superoxide scavengers such as Tiron and Tempo could block this conversion (Figure S5 in ESI Page S35), and inhibit the fluorogenic response in cell imaging experiments (Figure 2G). 3) Superoxide dose-dependently triggered on the fluorescence of probes F-Tz1-F-Tz4 (Figure 2D). 4) The probes were responsive to Paraquat or EGF treatment in live cells which are recognized to induce endogenous superoxide (Figure S40, S41 in Page S54, ESI). 4) NOX inhibitor DPI pre-treatment could abolish EGF-induced fluorogenic response of the probes (Figure S41). These results agreed with the recommendations presented in Nat Metab, 2022, 651 for testing probe selectivity, and should support the selectivity of the probes towards superoxide.

5. *It is highly recommended to report the shortcomings of using this method and how the*

authors could overcome. In addition, it should report the unavoidable defects in this method.

Thanks for raising this point.

Shortcomings of this work were discussed in the revised version (Page 21, text in blue). In brief, we have currently focused on the finding and verification of tetrazine-based probes for selectively imaging superoxide in live cells. Sufficient experiments were performed to support the selectivity of the probes, and the generality of tetrazine as a superoxide-selective trigger. However, detailed structure-sensitivity relationship study was not performed, and probe sensitivity could be enhanced further, which should be our future work.

6. EPR spin-trapping technique should be used for comparison purpose.

Thanks for raising this point.

We have performed EPR experiment to detect superoxide in HepG2 cells stimulated with H₂O₂ for comparison. Following literature instructions (J Clin Invest. 1993, 91, 46-52), we observed that the DMPO signals in response to superoxide increased as the cells were stimulated with increasing doses of H₂O₂, but the degree was not so dramatic (please check the figure below). Possible explanation for this result is that the facility in our institute is generally used to detect radicals in materials but not in biological samples which are aqueous solutions. It seemed that humidity had a lot impact on the sensitivity of the instrument.

On the other side, we agree with the reviewer that a comparison experiment should be carried out. We therefore alternatively detected superoxide in HepG2 cells by feeding the cells with DHE and then quantifying 2-hydroxyethidium by LCMS analysis, which is also a well-recognized method for specific detecting superoxide. The result showed similar trend with that observed by staining cells with probe F-Tz4. Detailed procedures for this experiment were described in Page S23, and the result was shown in Figure S44 in ESI (Page S56).

Following is the result of the EPR experiment.

Figure. EPR spectra of HepG2 cells treated with different concentration H_2O_2 . DMEM without cells was used as a positive control where in DMPO was used at 1 mM. Methods: HepG2 cells grown to confluence in a T 150 flask were treated with 0.05% trypsin for 3 min at 37°C. Medium with 10% FBS (5 ml) was added to the cells, which were then centrifuged at 1200 rpm for 3 min, washed twice with 5 ml of PBS. Cell counts were performed with a hemocytometer. Cell viability was assessed by exclusion of 0.2% trypan blue dye. DMEM without fetal bovine serum was added to adjust the cell concentration to $6 \times 10^6/\text{mL}$. The cells were exposed to different doses of H_2O_2 (0, 0.5, 1, 2 mM), together with DMPO (final concentration 1 mM). After an incubation time of 5 min, MeOH was added to the mixture (DMEM: MeOH = 1: 4) to decrease water proportion (Otherwise no signal would be detected). The mixed cell suspension was detected by EPR (Bruker A300, CenterField: 3507.00G, Sweep Width: 100G, Power: 20mW, Power Atten: 10dB, Frequency: 9.85 GHz, Modulation Amplitude: 2.00G, Modulation Frequency: 100.00 kHz).

7. A table of comparison with other fluorescent probes and other superoxide detection methods should be initiated and added to discussion section.

Thanks for this suggestion. Other fluorescent probes or methods for detecting superoxide have been discussed in the main text (Page 19, text in blue) and summarized in Table S4 in the ESI. For your quick reference, we also attached the table below.

Table S4. Summarization on various detection methods for superoxide ion.

Detection Method	Probe Structure	$\lambda_{\text{ex}}/\lambda_{\text{em}}$ (nm)	LOD	Application	Limitation	Ref
Mechanism 1 to identify superoxide ion: Proton abstraction						
Fluorescence and HPLC		510/595	ND	Detecting $\text{O}_2^{\cdot-}$ in vivo and in vitro	Intercalate into DNA and double-stranded RNA to cause interference; fluorescence of E^+ and 2-OH- E^+ overlaps; not	12
Fluorescence and HPLC		510/595	ND	Detecting $\text{O}_2^{\cdot-}$ in mitochondrial		12

Fluorescence and HPLC		566/638	ND		selective over ONOO ⁻ and ·OH; HE can increase the O ₂ ^{·-} dismutation rates toward H ₂ O ₂ , leading to an imprecise quantitative evaluation of O ₂ ^{·-}	13
Fluorescence and HPLC		479/597	ND			14
Two-Photon Fluorescence		485 or 800/559	1.63 nM	RAW264.7 treated with PMA	Not mentioned in the original report	15
Two-Photon Fluorescence		483 or 800/512	9.5 nM	Detecting O ₂ ^{·-} in mitochondrial	Not mentioned in the original report	16
Fluorescence		535-750/ 560-830	ND	Imaging O ₂ ^{·-} in cell and tissue,	Interference from hydroxyl radical	17
Mechanism 2 to identify superoxide ion: Caffeyl oxidation						
One-photon (OP) /two-photon (TP) imaging		491 or 800/515	2.3 nM	Reversible probe, imaging O ₂ ^{·-} levels in cell and mice with liver IR	Not mentioned in the original report	18
Two-Photon Fluorescence		370 or 800/495	21.5 nM	Imaging O ₂ ^{·-} in brains of mouse with depression	Not mentioned in the original report	19
Mechanism 3 to identify superoxide ion: Nucleophilic substitution						
Fluorescence		505/554	0.1 pM	Imaging O ₂ ^{·-} in human Jurkat T cells stimulated by butyric acid	Unstable in the physiological environmen (Fe ²⁺ , water, GSH)	20
Fluorescence		509/534	23 nM	Imaging O ₂ ^{·-} in intact live zebrafish embryos	Not mentioned in the original report	21
Two-Photon Fluorescence		365 or 800/500	1 nM	Deep-tissue imaging depth of -150 μm	Not mentioned in the original report	22
Fluorescence		600/730 or 790	10 μM	Ratiometric NIR fluorescent probe that real-time imaging O ₂ ^{·-} in liver of I/R mice injure	Not mentioned in the original report	23
Fluorescence		490/530	4.6 pM	Imaging superoxide in Macrophage	Unstable in the physiological environmen (Fe ²⁺ , water, GSH)	24

Mechanism 4 to identify superoxide ion : One-Electron Transfer						
UV-Vis	Nitro Blue Tetrazolium Reduction	NBT ⁺ 405nm/ MF ⁺ 530 nm	ND	Detecting O ₂ ^{•-} in solutions or cells containing NBT deposits	Potential artifacts; limited application scope	25
UV-Vis	Cytochrome c Reduction	550 nm	ND	Measure O ₂ ^{•-} generation by various enzymes, whole cells, and vascular tissues.	Limited application scope	26
Mechanism 5 to identify superoxide ion : Vibrational Spectroscopy						
Raman	-	-	ND	Detecting O ₂ ^{•-} and studying the related enzymes in biological systems.	The main limit of the Raman detection method is the extremely short lifetime of O ₂ ^{•-} species	27
Fourier Transform Infrared Spectroscopy.	-	-	ND	In situ analysis of adsorbed species and surface reactions.	Influential factors: different IR instruments, pH, or solvent	27
Electron Spin Resonance and Spin Trapping		-	ND	Detecting O ₂ ^{•-} in vitro and in vivo models	Expensive instrumentation, low sensitivity and selectivity with respect to other radicals present, low rate constants for spin trapping, low adduct stability, and lack of spin trap specificity	28, 29

ND: not determined

REVIEWER COMMENTS

Reviewer #1 (Remarks to the Author):

The revision is improved, and my original concerns have been addressed to some extent, but not satisfactorily.

The non-contractile nature of H9C2 cells and the generation of ROS by cardiac contraction must be acknowledged with appropriate refs. in the main manuscript.

Anesthesia as well as the duration of ischemia and reperfusion are essential for cardioprotection studies and must be presented in the main manuscript.

The concern on the use of anti-oxidants in cardioprotection must be explicitly acknowledged, given the serious problems in the translation of cardioprotection, see ref. 46.

Authors can not exclude hemodynamic side effects of coprostanone in mice by experiments in zebrafish embryos.

Authors must acknowledge the lack of causality between the Western on NFR 2, HO-1 and SOD2 and the observed cardioprotection in the main manuscript.

Reviewer #3 (Remarks to the Author):

The authors have addressed the comments satisfactorily

Reviewer #4 (Remarks to the Author):

This is a comprehensive and well-presented paper. It includes a sound scientific justification for using tetrazines as superoxide-based reactivity motifs, the merger of tetrazines into a palette of fluorescent probes with applications toward imaging endogenously produced superoxide, as well as the use of the technology to identify new superoxide-attenuating compounds.

My primary role is to evaluate whether the authors have addressed the concerns raised by Reviewer 2 in this resubmission.

Points 1-3 are suitably addressed.

Point 4: Thank you for including more information with regard the fluorescence quantification. Several questions remain:

a) No background subtraction was performed: did the authors confirm that the background signal is constant among experiments and does not substantially influence the quantification?

b) Please elaborate on the densitometry measurements: Were the cellular cytoplasmic regions outlined freehand and each cell fluorescence measured manually? Or were the images thresholded and batch processed? Which measurement function was used in ImageJ for the fluorescence intensity measurement (e.g., mean gray value, integrated density, etc.)? Are the measurements in the raw data for each condition pooled across the three experiments?

c) Were 18 cells / experimental condition measured, or were 18 cells per replicate measured? It is not uncommon to measure fluorescence intensities (even manually) from 100+ cells/condition to ensure the quantification is representative of the sample.

Points 5-9 are suitably addressed.

Point 10 is an important one and is largely addressed. However, DPI inhibition of NOX enzymes is not recommended (cf. the newly added Nat. Methods paper reference) as DPI is not specific to NOX. As per the recommendations of the Nat. Methods paper, more-specific NOX inhibitors should be used in place of DPI, or the lack of specificity of DPI should be noted.

Point 11 is suitably addressed.

Points 12-13 are suitably addressed.

Beyond the points above, there is inconsistent spectroscopic characterization of the synthetic intermediates. In some cases, only Rf and/or MS are provided as characterization data. In line with journal requirements, ^1H and ^{13}C NMR should be provided for organic compounds (<https://www.nature.com/ncomms/submit/chemical-characterisation>).

Reviewer #1 (Remarks to the Author):

The revision is improved, and my original concerns have been addressed to some extent, but not satisfactorily.

Response:

Thank you for your valuable feedback and suggestions. We have incorporated these suggestions into our revised paper by performing additional experiments to measure hemodynamics in a mouse model, citing all recommended references, and including limitations and experimental details of this study. We are deeply grateful for the reviewer's assistance in improving the quality of our manuscript.

The non-contractile nature of H9C2 cells and the generation of ROS by cardiac contraction must be acknowledged with appropriate refs. in the main manuscript.

Response:

Thanks for this suggestion. We have acknowledged this issue by adding the following statement to page 16, first paragraph.

“Although H9C2 cells are non-contractile and lack the ROS-generating property of normal cardiac contractility,⁴⁹⁻⁵¹ however, considering that the elevated ROS level during oxidative stress injury was much higher than that in physiological cardiac contractility, and the availability of cultured myocardial cells, we think a high-content screening model employing H9C2 cells should be acceptable; and further validation experiments can be performed in primary neonatal rat cardiomyocytes.”

Anesthesia as well as the duration of ischemia and reperfusion are essential for cardioprotection studies and must be presented in the main manuscript.

Response:

Thanks for this suggestion. In the revised manuscript, these essential details were added to the caption of Figure 6D.

“All the mice were anesthetized with tribromoethanol at a dose of 150 mg/kg before surgery. Mice myocardial ischemia and reperfusion injury were induced by the occlusion of the left anterior descending coronary artery for 45 min followed by 24 h reperfusion. The sham-operated group underwent the same procedure without ligation of the left anterior descending coronary artery.”

The concern on the use of anti-oxidants in cardioprotection must be explicitly acknowledged, given the serious problems in the translation of cardioprotection, see ref. 46.

Response:

Thanks for this suggestion. We have added a discussion on the use of antioxidants for cardioprotection in the first paragraph of page 16.

“It is important to note that many antioxidants exhibited therapeutic potential in preclinical studies but hardly achieved success in clinical trials. This discrepancy is presumably due to the ineffective scavenging of ROS, or their delayed administration at late reperfusion.^{4,48} However, this should not diminish the potential of antioxidants as cardioprotective agents. In this context, the search for effective superoxide modulators remains crucial for the development of cardioprotective agents.”

Authors can not exclude hemodynamic side effects of coprostanone in mice by experiments in zebrafish embryos.

Response:

Thanks for raising this point. We performed additional experiments to test the effects of coprostanone on hemodynamics in mice of sham operation or I/R injury. Mice were randomly divided into four groups, including sham, sham + 5 α Ch3 (100 mg/kg), I/R, and I/R + 5 α Ch3 (100 mg/kg) groups. After 3 days of pre-treatment with 5 α Ch3 or vehicle, ischemia/reperfusion and sham operation were performed. After 45 min ischemia followed by 24 h reperfusion, each anesthetized mouse was micro-cannulated with a 1.4F microcatheter (Millar Instrument Inc, USA). Key parameters, including left ventricular systolic pressure (LVSP), left ventricular end-diastolic pressure (LVEDP) and the maximum and minimum rates of left ventricular pressure changes (dp/dt max and dp/dt min), were automatically recorded.

Compared with the sham-operated group, the abnormal values of LVSP and dp/dt max were observed in the mice subjected to I/R injury. The pretreatment of 5 α Ch3 (100 mg/kg) effectively improved these parameters in mice subjected to I/R injury; while it caused no significant effects on these parameters in the sham group. Apart from eliminating the side effects of 5 α Ch3 on the hemodynamics of sham-operated mice, these data highlighted the beneficial effects of 5 α Ch3 against the impaired hemodynamic parameters of I/R mice.

Detailed experimental procedures were described in the section “Testing the effects of coprostanone on hemodynamics in mice” in ESI. Results were described on page 18 in the main text and Figure S59 in ESI.

Figure S59. Hemodynamic parameters in mice subjected to cardiac I/R injury with 5 α Ch3 pretreatment. (A) LVSP, left ventricular systolic pressure; (B) LVEDP, left ventricular end diastolic pressure; (C) dp/dt max: the maximum rate of left ventricular pressure change; (D) dp/dt min: the minimum rate of left ventricular pressure change (n = 5-7). Data were expressed as mean \pm SD and analyzed with one-way ANOVA. *P<0.05, ** P < 0.01 vs. I/R group.

Authors must acknowledge the lack of causality between the Western on NFR 2, HO-1 and SOD2 and the observed cardioprotection in the main manuscript.

Response:

Thanks for your feedback. We have made a revision to address this point by adding the following statement in the last paragraph of page 18.

“While the exact mechanism by which coprostanone induces cardioprotection remains to be explored, its effect to inhibit superoxide overload has been established, and the enhancement of the Nrf2-HO-1/SOD2 signaling pathway may play a role in this effect.”

Reviewer #4 (Remarks to the Author):

This is a comprehensive and well-presented paper. It includes a sound scientific justification for using tetrazines as superoxide-based reactivity motifs, the merger of tetrazines into a palette of fluorescent probes with applications toward imaging endogenously produced superoxide, as well as the use of the technology to identify new superoxide-attenuating compounds.

My primary role is to evaluate whether the authors have addressed the concerns raised by Reviewer 2 in this resubmission.

Points 1-3 are suitably addressed.

Points 5-9 are suitably addressed.

Point 11 is suitably addressed.

Points 12-13 are suitably addressed.

Response:

We are grateful for the reviewer's comments.

Point 4: Thank you for including more information with regard the fluorescence quantification. Several questions remain:

a) No background subtraction was performed: did the authors confirm that the background signal is constant among experiments and does not substantially influence the quantification?

Response:

Thank you for bringing this to our attention. To ensure consistency, the same microscope parameters were used across all conditions within a single experiment. This approach helped to maintain constant background signals. As the background fluorescence was constant and insignificant, it had no significant impact on the quantification results.

b) Please elaborate on the densitometry measurements: Were the cellular cytoplasmic regions outlined freehand and each cell fluorescence measured manually? Or were the images thresholded and batch processed? Which measurement function was used in ImageJ for the fluorescence intensity measurement (e.g., mean gray value, integrated density, etc.)? Are the measurements in the raw data for each condition pooled across the three experiments?

Response:

Thanks for raising this point. We have outlined the cellular cytoplasmic regions freehand and manually measured the fluorescence intensity of each cell using the mean gray values. No threshold was used. The

raw data measurements were pooled across various conditions and three parallels within one experiment. This information was added in the “Fluorescence confocal imaging” section in ESI.

c) Were 18 cells / experimental condition measured, or were 18 cells per replicate measured? It is not uncommon to measure fluorescence intensities (even manually) from 100+ cells/condition to ensure the quantification is representative of the sample.

Response:

Thanks for bringing this to our attention. We are sorry for any confusion. We previously randomly selected cells for quantification as the cellular fluorescence brightness among replicates of the same condition was similar. However, following the reviewer’s suggestion of quantifying more cells, we have re-performed the analysis, measuring a larger number of cells. Specifically, the number of cells measured per condition was in the range of 40-120.

Point 10 is an important one and is largely addressed. However, DPI inhibition of NOX enzymes is not recommended (cf. the newly added Nat. Methods paper reference) as DPI is not specific to NOX. As per the recommendations of the Nat. Methods paper, more-specific NOX inhibitors should be used in place of DPI, or the lack of specificity of DPI should be noted.

Response:

Thanks for raising this point. We have reviewed the relevant literature and recognized the lack of specificity of DPI for NOX. To address this limitation, we conducted an additional experiment using VAS2870, a more specific inhibitor of NOX. Our results showed that pretreatment of A549 cells with VAS2870 (at concentrations of 10 and 20 μ M) for 1 hour dose-dependently reduced the cellular probe fluorescence response upon stimulation with EGF. We have included this new data in Figure S41 and acknowledged the unspecificity of DPI and the use of VAS2870 in the main text, with appropriate citations of the relevant literature.

Figure S41. F-Tz4 imaging of endogenous superoxide in live A549 cells stimulated by EGF. A) Confocal microscopy images of A549 cells first treated with 0.5 $\mu\text{g/mL}$ EGF for 30 min, then stained with F-Tz4 (5 μM) for 30 min before imaging (Scale bar: 50 μm). The cells were either pretreated with DPI (at a concentration of 5 μM) for 30 min prior to EGF treatment or left untreated as a control. B) Confocal microscopy images of A549 cells treated with 0.5 $\mu\text{g/mL}$ EGF for 30 min, then stained with F-Tz4 (5 μM) for 30 min before imaging (Scale bar: 25 μm). The cells were either pretreated with VAS2870 (at concentrations of 10 and 20 μM) for 60 min prior to EGF treatment or left untreated as a control. C-D) Quantified relative mean fluorescence intensities of the cells. **P < 0.01, ***P < 0.001; versus untreated cells. **P < 0.01, ***P < 0.001; versus untreated cells.

Beyond the points above, there is inconsistent spectroscopic characterization of the synthetic intermediates. In some cases, only Rf and/or MS are provided as characterization data. In line with journal requirements, ^1H and ^{13}C NMR should be provided for organic compounds (<https://www.nature.com/ncomms/submit/chemical-characterisation>).

Response:

Thank you for bringing this to our attention. We acknowledge that we had previously neglected to include NMR data for some intermediates with known structures. We have rectified this oversight in this revision by conducting additional NMR analysis. To provide easy reference, we have summarized the current state of characterization data for all intermediates and final products in the paper. With the exception of those intermediates that were unstable or used directly in the next step without purification, all are now reported with NMR and mass spectrometry data.

Comp	New Comp (Yes/No)	^1H -NMR	^{13}C -NMR	MS ^[a]	Rf Value	Note or Ref
Final products						
F-Tz1	Yes	☑	☑	☑	☑	-
F-Tz2	Yes	☑	☑	☑	☑	-
F-Tz3	Yes	☑	☑	☑	☑	-
F-Tz4	Yes	☑	☑	☑	☑	-
F-Tz5	Yes	☑	☑	☑	☑	-
F-Tz6	Yes	☑	☑	☑	☑	-
F-Tz7	Yes	☑	☑	☑	☑	-
F-Tz8	Yes	☑	☑	☑	☑	-
F-Tz9	No	☑	☑	☑	☑	10.1002/anie.202112931
Tz1	No	☑	☑	☑	☑	10.1016/j.tetlet.2014.07.012 (2014)
Tz2	No	☑	☑	☑	☑	10.1016/j.tetlet.2014.07.012 (2014)
Tz3	No	☑	☑	n.s.	☑	10.1016/j.tetlet.2014.07.012 (2014)
Tz4	No	☑	☑	☑	☑	10.1021/jacs.9b08677
O1	No	☑	☑	☑	☑	10.1016/j.tetlet.2010.06.139 (2010)
Intermediates						
S1	No	☑	☑	☑	☑	10.1016/j.bioorg.2022.106069
S2	No	☑	☑	n.s.	☑	10.1016/j.bmc.2016.09.041
S3	Yes	-	-	☑	☑	Directly used without purification

S4	Yes	[x]	[x]	[x]	[x]	-
S5	No	-	-	[x]	[x]	Directly used without purification
S6	No	[x]	[x]	[x]	[x]	10.1021/acs.orglett.2c02596
S7	Yes	[x]	[x]	[x]	[x]	-
S8	Yes	[x]	[x]	[x]	[x]	-
S9	Yes	-	-	[x]	-	unstable intermediate
S11	No	[x]	[x]	[x]	[x]	10.1002/chem.201801701
S13	No	-	-	[x]	[x]	Directly used without purification
S15	No	[x]	[x]	[x]	[x]	10.1039/c0nj00578a
S16	No	[x]	[x]	[x]	[x]	
S18	No	[x]	-	n.s.	[x]	10.1021/acs.jmedchem.5b01249
S19	No	[x]	-	n.s.	[x]	10.1021/jo200853j
S20	No	[x]	[x]	[x]	[x]	10.1002/anie.202112931

n.s.: No mass signal was observed probably due to poor ionization.

REVIEWERS' COMMENTS

Reviewer #1 (Remarks to the Author):

All my concerns have now been adequately addressed.

Reviewer #4 (Remarks to the Author):

The authors have addressed my concerns on their revision, and I am happy to recommend its acceptance. I congratulate the authors on such an important advance for superoxide imaging.

Reviewer #1 (Remarks to the Author):

All my concerns have now been adequately addressed.

Response: We appreciate the positive comment from the reviewer.

Reviewer #4 (Remarks to the Author):

The authors have addressed my concerns on their revision, and I am happy to recommend its acceptance. I congratulate the authors on such an important advance for superoxide imaging.

Response: We appreciate the positive comment from the reviewer.